# Inflammation, Oxidative Stress, and Endothelial Dysfunction in the Pathogenesis of Vascular Damage: Unraveling Novel Cardiovascular Risk Factors in Fabry Disease

**DOI:** 10.3390/ijms25158273

**Published:** 2024-07-29

**Authors:** Denise Cristiana Faro, Francesco Lorenzo Di Pino, Ines Paola Monte

**Affiliations:** Department of General Surgery and Medical-Surgical Specialties (CHIRMED), University of Catania, Via S. Sofia 78, 95100 Catania, Italy; denisefaro88@gmail.com (D.C.F.); francesco.dipino@icloud.com (F.L.D.P.)

**Keywords:** Anderson Fabry disease, x-linked transmission, inflammation, cardiac hypertrophy, endothelial dysfunction

## Abstract

Anderson-Fabry disease (AFD), a genetic disorder caused by mutations in the α-galactosidase-A *(GLA)* gene, disrupts lysosomal function, leading to vascular complications. The accumulation of globotriaosylceramide (Gb3) in arterial walls triggers upregulation of adhesion molecules, decreases endothelial nitric oxide synthesis, and induces reactive oxygen species production. This cascade results in fibrotic thickening, endothelial dysfunction, hypercontractility, vasospasm, and a pro-thrombotic phenotype. AFD patients display increased intima-media thickness (IMT) and reduced flow-mediated dilation (FMD), indicating heightened cardiovascular risk. Nailfold capillaroscopy (NFC) shows promise in diagnosing and monitoring microcirculatory disorders in AFD, though it remains underexplored. Morphological evidence of AFD as a storage disorder can be demonstrated through electron microscopy and immunodetection of Gb3. Secondary pathophysiological disturbances at cellular, tissue, and organ levels contribute to the clinical manifestations, with prominent lysosomal inclusions observed in vascular, cardiac, renal, and neuronal cells. Chronic accumulation of Gb3 represents a state of ongoing toxicity, leading to increased cell turnover, particularly in vascular endothelial cells. AFD-related vascular pathology includes increased renin-angiotensin system activation, endothelial dysfunction, and smooth muscle cell proliferation, resulting in IMT increase. Furthermore, microvascular alterations, such as atypical capillaries observed through NFC, suggest early microvascular involvement. This review aims to unravel the complex interplay between inflammation, oxidative stress, and endothelial dysfunction in AFD, highlighting the potential connections between metabolic disturbances, oxidative stress, inflammation, and fibrosis in vascular and cardiac complications. By exploring novel cardiovascular risk factors and potential diagnostic tools, we can advance our understanding of these mechanisms, which extend beyond sphingolipid accumulation to include other significant contributors to disease pathogenesis. This comprehensive approach can pave the way for innovative therapeutic strategies and improved patient outcomes.

## 1. Introduction: Anderson-Fabry Disease Overview

Anderson-Fabry disease (AFD) is an X-linked lysosomal storage disorder caused by pathogenic variants in the α-galactosidase A gene (α-GAL-A, *GLA* gene), resulting in reduced or absent lysosomal enzyme activity and impaired lysosomal function [1]. It is a monogenic, X-linked recessive hereditary disease with an estimated birth prevalence of 1:40,000–117,000 [2,3]. Hemizygous males typically exhibit full disease features due to almost zero or markedly reduced α-galactosidase A activity, presenting with neurological, dermatological, renal, cardiovascular, cochleo-vestibular, and cerebrovascular symptoms. Heterozygous females can have variable expressivity, ranging from very mild to severe [1,4]. Once considered mere carriers, they actually exhibit a broad spectrum of disease manifestations due to lyonization, leading to cellular mosaicism [5]. X-chromosome inactivation occurs randomly in females, balancing gene dosage between females with two X chromosomes and males with one. Females expressing the wild-type X chromosome allele may have few symptoms, while those expressing the mutated allele can exhibit severe disease like hemizygous males [6,7].

AFD is a multisystemic disorder that begins with cellular dysfunction, followed by a cascade of functional impairment in various organs and culminating in structural damage that progresses over years or even decades, leading to late complications and renal, cardiac, or cerebrovascular failure. The progressive organ damage is closely linked to the involvement of vascular endothelial and smooth muscle cells, particularly in the microcirculation, pericytes, cardiomyocytes, various epithelial cells in the kidney, and neuronal cell types in the central and peripheral nervous systems [4,8]. The accumulation of metabolites may trigger secondary pathological processes, including inflammation, ischemia, hypertrophy, and fibrosis [9,10].

There is significant inter- and intrafamilial variability in the age of onset, clinical features, and disease progression [5]. The primary pathological process begins in childhood or even fetal development [11]. However, unlike many other lysosomal storage diseases, most patients remain clinically asymptomatic in the early years [12]. 

### 1.1. Clinical Phenotypes

Biochemically, AFD presents two distinct phenotypes: classic and non-classic or late-onset. The prevalence of the classic AFD phenotype in male patients with the *GLA* gene mutation variant is 0.12%, or 1 in 1000 screened individuals [13].

The classic form of AFD is commonly found in male patients who exhibit almost absent or very low levels of α-Gal A enzyme activity (<1%). This leads to early onset of symptoms and significant accumulation of Gb3 in various cell types. This complex scenario is associated with an increased risk of multiorgan failure and premature mortality. The most common presenting signs in classically affected hemizygous males include burning pain at the extremities, hypo- or hyperhidrosis, transient ischemic attacks, stroke, cutaneous angiokeratomas, proteinuria, cardiomyopathy, cardiac arrhythmias, cochleo-vestibular, and gastrointestinal disturbances. Generally, significant cardiac and renal complications develop after the age of 20 years old.

Patients with “non-classical” or “late-onset“ disease possess residual α-galactosidase A activity ranging between 2 and 20% of normal. They often exhibit a milder form of the disease, with symptoms typically manifesting in the fourth to sixth decade of life, often limited to a single organ system, for example, the heart, with the extent of involvement related to the type of mutation (such as N215S), the degree of residual enzyme activity, and the levels of lyso-Gb3 in the tissues [14]. The cardiac variant is the most widely reported atypical form, presenting with hypertrophic cardiomyopathy (HCM), arrhythmias, and myocardial infarction without coronary artery significant plaques. In AFD, the accumulation of lyso-Gb3 in cardiomyocytes, coupled with the activation of a complex chronic inflammatory pathway, leads to progressive left ventricular hypertrophy (LVH) and heart failure with preserved ejection fraction (HFpEF). Recent studies suggest that AFD should be considered in all cases of concentric, unexplained etiology, after excluding the more common causes [15].

Additionally, fibrosis and involvement of conduction tissue, along with altered electrical properties of cardiomyocytes (including ion channel expression and/or membrane trafficking), result in the development of ventricular arrhythmias and conduction disturbances [16,17].

### 1.2. Genetic Variants and Genotype-Phenotype Correlations

The Human Gene Mutation Database (http://www.hgmd.cf.ac.uk/ac/gene.php?gene=GLA, accessed on 3 May 2024) currently reports more than 900 variants of the *GLA* gene, approximately 75% of which are point mutations. Genetic variants are classified into categories: pathogenic, likely pathogenic, VUS (variants of uncertain significance), likely benign, and benign, as per the American College of Medical Genetics and Genomics guidelines. Most of these mutations are classified as pathogenic [18].

The diagnosis of AFD involves a comprehensive clinical, biochemical, and genetic evaluation due to the complex nature of the disorder. The genotype-phenotype correlation in AFD is complicated due to the disease’s rarity, allelic heterogeneity, variation in clinical expressivity, and lack of published clinical data [19,20]. Although progress has been made in understanding the natural history of AFD, a better understanding of genotype-phenotype relationships is needed for predicting disease progression, setting therapeutic outcome expectations, and assisting in treatment selection [21].

### 1.3. Gender Differences and the Role of Sex Hormones in the Severity of Fabry Disease

Phenotypic diversity in female AFD patients is primarily due to genetic factors, including XCI and epigenetic modifications [22,23]. The variability in female AFD disease severity is linked to X-chromosome inactivation (XCI), where one of the two X chromosomes in females is randomly inactivated, creating a mosaic of affected cells. Females typically carry heterozygous *GLA* gene mutations, with a 50% chance of passing the defective gene to offspring [23,24].

XCI can cause varied α-galactosidase A activity, leading to diverse symptoms that appear later and progress more slowly in females compared with males [20,25,26]. Female AFD patients show clinical pictures, influenced by XCI skewness favoring the mutant allele [27,28]. Some studies correlate AFD severity with XCI patterns, while others do not [27,28]. Additional factors, such as allele-specific DNA methylation at the GLA promoter, regulate gene expression, contribute to phenotype variability, and may influence disease expression, severity, and lysoGb3 accumulation.

While sex hormones’ impact on AFD severity is hypothesized, it remains unproven. Estrogens might offer vascular protection to premenopausal women, but this effect is not definitively established. Elevated growth factors like VEGF-A (Vascular endothelial growth factor A) and FGF2 (Fibroblast growth factor 2) in female patients suggest gender-specific mechanisms [29].

Recent findings highlight proteins involved in inflammation and coagulation/fibrinolysis, such as ANT3 antibodies, HRG (Histidine-rich glycoprotein), FINC (fibronectin), and plasminogen, in women [30]. In men, the protein 14-3-3 zeta is significant. Both sexes show complement system activation, indicated by downregulation of C1QB (B-chain polypeptide of serum complement subcomponent C1q) and CO5 (Complement Component 5). Vitronectin expression is downregulated in male AFD patients and in female patients with complications, linking it to atherosclerotic cardiovascular disease pathogenesis, and serving as a biomarker for AFD progression. Understanding these mechanisms can aid in developing targeted therapies and improving patient outcomes.

### 1.4. Therapy: A Brief Overview

AFD is a progressive and potentially life-threatening lysosomal storage disorder whose natural course can be significantly altered by enzyme replacement therapy (ERT). Thus, early diagnosis and prompt management are the optimal goals for these patients, yet they pose a challenge. ERT was introduced in Europe in 2001 and in the United States in 2003. Recombinant human α-galactosidase A is commercially available in two forms: agalsidase alpha and agalsidase beta [31,32]. 

Evidence accumulated since the introduction of ERT has confirmed its ability to prevent or at least halt disease progression, with its effectiveness primarily depending on the timeliness of initiation [33,34]. A reduction in severe cardiac events has been observed in previous studies. Specifically, ERT has been shown to reduce myocardial accumulation of Gb3, and early treatment has resulted in a reduction in left ventricular mass and an improvement in intracardiac conduction, preventing fibrosis development [35].

Since 2016, chaperone therapy with migalastat has been available, capable of stabilizing the residual endogenous enzyme in the serum, which is present but not bioavailable due to misfolding and inability to enter the cell. By correcting the enzyme’s misfolding, migalastat has proven to restore its capacity to enter the cell via a chaperone mechanism and to reestablish the degradation function of glycolipids. This orally administered drug is therefore indicated in patients with missense mutations identified as susceptible (“amenable”). Available evidence to date has shown that the drug is safe and well-tolerated, and its efficacy is largely comparable to ERT [36,37].

## 2. Pathogenesis of Organ Complications Related to Vascular Damage from Gb3 Accumulation and Inflammation

AFD can be demonstrated as a storage disorder through electron microscopy of affected tissues and specific in situ immunodetection of Gb3 using monoclonal antibodies [8]. While Gb3 accumulation is the primary pathogenic event, the subsequent secondary disturbances at cellular, tissue, and organ levels that lead to clinical manifestations are not fully understood [4,38]. Significant abnormalities include lysosomal inclusions or lipid deposits in various cell types, prominently in vascular endothelial and smooth muscle cells, cardiac cells, renal epithelial cells, and nerve cells, including dorsal root ganglia and some central nervous system neurons [38]. Chronic Gb3 accumulation prompts a state of chronic toxicity. Although there is no evidence of massive cell death, increased turnover of certain cells, such as vascular endothelial cells, is likely.

AFD is indeed largely a systemic vascular disease-causing cardiovascular complications and cerebral ischemia. Several mechanisms contribute to ischemic tissue damage, including occlusion and luminal obstruction due to Gb3 accumulation in vascular endothelial cells, disruption of the balance between vasodilators and vasoconstrictors, and thromboembolic complications [39].

Early peripheral neuropathy reflects functional deterioration of neuronal cells in the autonomic and somatosensory peripheral nervous systems due to Gb3 deposits in the vasa vasorum of small myelinated and unmyelinated fibers. Numerous studies have demonstrated cerebral circulation dysfunction in Fabry patients, particularly significant hyperperfusion in the posterior cerebral circulation [40]. This hyperperfusion is not ubiquitous, suggesting heterogeneity in response to glycolipid accumulation in different vascular beds. Calcifications indicating end-stage organ damage are demonstrable in hyperperfused brain regions such as white matter and posterior thalamic regions [41].

Hypohidrosis is a sign of selective damage to peripheral nerves [42], although it has also been attributed to lipid deposits in small vessels surrounding sweat glands. Vascular skin lesions (angiokeratomas) are caused by the weakening of capillary walls due to the accumulation of Gb3 and the development of vascular ectasias in the dermis and epidermis. Early gastrointestinal manifestations are due to Gb3 accumulation in the vascular endothelium of mesenteric blood vessels, unmyelinated neurons, perineural cells, and autonomic ganglia in the gastrointestinal tract [43].

The pathogenesis of the disease in the heart and kidneys is not well understood. The characteristic hypertrophic cardiomyopathy of AFD is associated with increased (IMT)-media thickness, like that observed in common cardiovascular diseases. In Fabry patients’ plasma, there is an increased proliferative response of vascular smooth muscle cells (SMCs) in culture compared with control plasma, suggesting that a circulating factor is partly responsible for cardiac hypertrophy in this disease [44], a phenomenon similar to that observed in vascular endothelial cells [45].

About the mechanism of renal failure in AFD, beyond the accumulation of material, pathological alterations and the presence of progressive glomerulosclerosis and proteinuria resemble other proteinuric nephropathies, such as those occurring in diabetes mellitus [46]. Fabry proteinuria does not respond to enzyme replacement infusions, but if the analogy with diabetes is correct, it may be associated with overexpression of cathepsin-L, as well as altered dynamin processing, and thus may be reversible with appropriate treatment [47].

### 2.1. Clinical Studies in Humans: Increased IMT at Doppler Imaging and Capillary Anomalies

As discussed above, various studies have evaluated cardiovascular involvement in patients with AFD mutations. However, there is currently limited data on early cardiovascular damage in AFD without cardiac hypertrophy. To expand knowledge of the early signs of cardiovascular involvement, a study conducted by Costanzo and Monte et al. [48] aimed to comprehensively evaluate cardiac, macrovascular, and microvascular functions in Fabry mutation carriers without cardiac hypertrophy. They found that Tissue Doppler Imaging parameters and longitudinal deformation were able to show preclinical cardiac function impairments in mutation carriers; the same patients also had more pronounced early macrovascular involvement, assessed by IMT and FMD (Flow-Mediated Dilatation), and more often microvascular alterations, assessed through capillaroscopy.

These results suggest that in Fabry mutation carriers, early myocardial function anomalies with involvement of macrovascular and microvascular systems can be detected before the development of cardiac hypertrophy.

It has also been demonstrated that these patients had increased IMT in the carotids in the absence of plaques and reduced FMD of the brachial arteries, confirming previous findings [47,49].

Glycosphingolipid deposition in the vascular wall may explain the increased vascular IMT observed in AFD. A predominant accumulation of Gb3 in the intima and smooth muscle of the media of arterial walls has been demonstrated, which can lead to thickening of the extracellular matrix and calcifications. Moreover, glycosphingolipids have a particular affinity for the vascular endothelium. 

Additionally, a significantly higher number of capillaroscopic alterations were observed. Most patients with AFD, in fact, showed atypical capillaries [48]. 

A possible explanation for the capillaroscopic alterations could be altered endothelial function due to intracellular glycosphingolipid accumulation [44]. Stemper and Hilz et al. [50] also postulated an alteration in the perfusion of end organs, most likely with arterio-venous shunting and inadequate perfusion of capillaries. This could lead not only to structural injuries at the level of shunts but also to impaired sympathetic vasomotor control due to small fiber neuropathy [51]. Moreover, a significant physiological control of smooth muscle activity is indirectly mediated by endothelial cells with the release of vasodilating prostaglandins (prostacyclin) and potent vasoconstrictors such as endothelin-1. In AFD, sphingolipid accumulation can cause an imbalance of these regulatory mechanisms. The clinical significance of these findings is not clear; however, a high number of capillaroscopic alterations may suggest early microvascular involvement in AFD before the development of macrovascular damage [48] (Table 1).

### 2.2. Effects of Gb3 on Endothelial Health and Smooth Muscle Cell Growth in Fabry Disease: Evidence from In Vitro Studies

In vitro studies consistently show that Gb3 exerts harmful effects on endothelial and SMCs, contributing to vascular complications in Fabry disease. The accumulation of Gb3 in endothelial cells markedly increases the production of reactive oxygen species (ROS), impairing nitric oxide (NO) synthesis. This oxidative stress disrupts NO signaling and inhibits endothelial NO synthase (eNOS) activity, resulting in reduced NO bioavailability and endothelial dysfunction [45,58]. Furthermore, evidence suggests that the widespread intracellular accumulation of Gb3 disrupts other cellular components regulating endothelial function [59]. Other studies reveal that total eNOS and phosphorylated eNOS, both membrane-associated, are downregulated in Gb3-loaded human endothelial cells (ECs), while iNOS (inducible nitric oxide synthase) expression is upregulated. This dysregulation of the L-arginine/NO pathways may adversely affect vascular function and blood flow to vital organs, contributing to the risk of vasculopathy and cardiovascular complications [60]. Additionally, Gb3 has been shown to upregulate the expression of cyclooxygenase-2 (COX-2), a pro-inflammatory enzyme that exacerbates endothelial inflammation. This upregulation is linked to enhanced ROS production, creating a feedback loop that further damages endothelial function. COX-2-derived prostaglandins may play a compensatory role for decreased NO bioavailability, potentially explaining some cardiovascular effects associated with COX-2 inhibitors [58,61]. Experiments comparing Gb3 accumulation to GLA deficiency in human macro- and microvascular cardiac ECs (HMiVECs) show that Gb3 loading disrupts several key endothelial pathways, such as eNOS, iNOS, COX-1, and COX-2, while *GLA* silencing shows no effects, and that microvascular ECs are more susceptible to Gb3 loading than macrovascular ECs [60]. These data indicate that endothelial dysfunction in Fabry disease is primarily due to Gb3 accumulation rather than *GLA* deficiency. Gb3 accumulation affects microvascular endothelial cells more severely, altering their phenotype to a vasoconstrictive and pro-inflammatory state, consistent with previous observations that atherosclerosis is not a common finding in Fabry disease [62]. Gb3 enhances COX-2 expression in HMiVECs, while COX-1 expression remains unaffected, suggesting that vasoconstrictive COX-2 derivatives contribute to microvascular dysfunction in Fabry disease [63,64]. Moreover, Gb3 upregulates VCAM-1 (Vascular cell adhesion protein 1) but not ICAM-1 (Intercellular Adhesion Molecule 1) expression in HMiVECs, indicating a pro-inflammatory phenotype in Fabry disease. Increased plasma levels of other endothelial atherosclerotic factors and leukocyte adhesion molecules further support these findings [45,65]. The inflammatory environment induced by Gb3 promotes adhesion molecule expression, such as ICAM-1 and VCAM-1, facilitating leukocyte adhesion and migration to the endothelium. This enhances vascular inflammation and oxidative stress. Studies have shown that Gb3-treated endothelial cells exhibit significantly higher levels of these adhesion molecules, which are critical in the pathogenesis of vascular complications in Fabry disease [45]. Additionally, Gb3 stimulates vascular SMCs proliferation via signaling pathways, including TGF-β and NF-κB, leading to vascular remodeling and stiffness [44].The activation of these pathways leads to increased SMCs proliferation, contributing to vascular remodeling and stiffness. Hwang et al. (2023) demonstrated that Gb3 also induces autophagy-dependent regulation of necroptosis in endothelial cells, indirectly supporting SMCs growth and vascular remodeling [66]. A novel mechanism by which Gb3 impairs endothelial function involves the degradation of calcium-activated potassium channel 3.1 (KCa3.1) through clathrin-mediated endocytosis and lysosomal degradation, disrupting calcium signaling and NO synthesis [67]. Gb3 also induces various angiogenesis factors, such as VEGF, VEGFR2, TGF-β, and FGF-2, in endothelial and smooth muscle cells, contributing to vascular dysfunction and angiogenesis in AFD. This was observed in both in vitro and in vivo models, indicating its role in vascular dysfunction and angiogenesis in AFD [45,60,64,68,69,70,71,72]. Research by De Francesco et al. shows that PBMCs (peripheral blood mononuclear cells) from Fabry patients exhibit higher proinflammatory cytokine expression and production when cultured, mediated by TLR4 (Toll-like receptor 4), indicating a proinflammatory response through TLR4 activation [73]. Pollmann et al. highlight that endothelial dysfunction in Fabry disease is linked to glycocalyx degradation, with AGAL-deficient endothelial cells showing reduced glycocalyx and increased monocyte adhesion. Increased angiopoietin-2, heparanase, and NF-κB (Nuclear Factor kappa B) expression were improved with ERT and anti-inflammatory drugs [74].

### 2.3. Immune Cells Involved in Fabry Disease and Their Interplay with Endothelium and Inflammation

The accumulation of Gb3 in various tissues results in significant endothelial dysfunction and inflammation, where the immune system plays a critical role. Neutrophils have a leading role in AFD-related inflammation. These cells produce myeloperoxidase (MPO), an enzyme that is significantly elevated in AFD patients and is closely associated with endothelial dysfunction and increased cardiovascular risk. Elevated MPO levels indicate a state of neutrophil activation, which generates ROS, exacerbating oxidative stress and endothelial damage. This oxidative stress contributes to a vicious cycle of inflammation and vascular injury [75]. Monocytes and macrophages are essential to the inflammatory response in AFD. Monocytes differentiate into macrophages, which produce various pro-inflammatory cytokines and enzymes, including MPO. These macrophages accumulate in tissues where Gb3 is deposited, perpetuating chronic inflammation and tissue remodeling. The persistent activation of these cells leads to enhanced inflammatory responses and further endothelial damage [76,77]. T cells, particularly the Th1 and Th17 subsets, contribute to the immune response in AFD. Th1 cells produce interferon-gamma, while Th17 cells secrete IL-17, both of which promote inflammation. These cytokines are implicated in the vascular dysfunction observed in AFD, suggesting that T cells play a significant role in sustaining chronic inflammation and endothelial injury [77]. One of the first responses is the upregulation of adhesion molecules such as VCAM-1 and ICAM-1 on the surface of endothelial cells. These molecules facilitate the adhesion and transmigration of immune cells into the vascular endothelium, promoting further inflammation and endothelial damage [45,76]. Endothelial activation in AFD is characterized by a shift towards a pro-inflammatory and pro-thrombotic state. Activated endothelial cells secrete cytokines and chemokines, attracting more immune cells and perpetuating the cycle of inflammation. This chronic inflammatory state leads to vascular remodeling, including intimal hyperplasia and fibrosis, contributing to the narrowing of blood vessels and impaired blood flow [77]. Oxidative stress plays a crucial role in AFD-related endothelial dysfunction. MPO released by neutrophils generates ROS, which cause direct oxidative damage to endothelial cells. This oxidative stress not only damages the endothelium but also amplifies the inflammatory response, creating a self-perpetuating cycle of vascular pathology.

### 2.4. Molecular Pathways Involved in Ischemia, Prothrombotic State, and ROS Production (Figure 1)

A deeper look into the pathophysiology of organ damage in AFD reveals the involvement of multiple molecular pathways, including ischemia, prothrombotic state, and ROS production. AFD-induced vasculopathy is characterized by abnormalities in blood components, blood flow, and the vascular wall, significantly contributing to vascular dysfunction in line with Virchow’s triad (see Figure 1). Elevated levels of soluble ICAM-1, soluble VCAM-1, P-selectin, and plasminogen activator inhibitor (PAI), along with reduced thrombomodulin and heightened CD11b monocyte expression, underscore a prothrombotic state in AFD [38]. Excessive ROS production, potentially stemming from glycolipid accumulation affecting endothelial function, exacerbates the condition [45].

Impaired α-GAL-A activity results in widespread intralysosomal accumulation of Gb3 and other minor compounds, primarily in the kidneys, from fetal life [16], though patients are asymptomatic in early life [78,79]. Elevated levels of deacylated lyso-Gb3 in Fabry patients inhibit α-galactosidase A and B and promote SMCs proliferation, contributing to increased IMT. Lyso-Gb3 accumulates in the plasma [80] and urine [81] of Fabry patients, contributing to the disease’s metabolic load and leading to structural damage and abnormal cellular functions like impaired muscle cell contractility and altered surface molecule expression [82]. The primary vascular pathology in AFD leads to severe organ complications, including hypertrophic cardiomyopathy, stroke, and chronic kidney disease [80,83,84,85]. These metabolites may trigger secondary pathological processes with systemic effects, including inflammation, ischemia, oxidative stress, altered immune responses, hypertrophy, and fibrosis, which are pivotal in the disease’s clinical manifestations [4,78]. 

Progressive organ damage in AFD involves vascular endothelial and smooth muscle cells, particularly in the microcirculation, pericytes, cardiomyocytes, various kidney epithelial cells, and neuronal cells in the central and peripheral nervous systems [4,8]. The severity of tissue damage correlates with residual enzyme activity and the metabolic load of the disease, defined by accumulated substrates [78]. Despite the systemic nature of the deficit, organ susceptibility varies, with some organs, like the liver, being resistant while others, such as the heart, are highly susceptible.

Disease progression affects multiple systems over time: both primary and secondary processes progressively damage organ systems, contributing to the multisystem failure and organ fragility characteristic of AFD and potentially leading to late complications and failures in renal, cardiac, or cerebrovascular functions. Ischemia, particularly affecting small vessels in the cerebrovascular system, heart, kidneys, peripheral nervous system, and skin, plays a crucial role in the disease’s phenotype, reflecting its systemic vasculopathy.

### 2.5. Role of Oxidative Stress

Advanced Oxidation Protein Products are significantly higher in AFD patients compared with controls, with decreased antioxidant defenses such as ferric reducing antioxidant power (FRAP) and thiol groups [86]. In treatment-naïve subjects with AFD-related mutations, altered oxidative stress parameters and early signs of organ damage were observed despite normal lyso-Gb3 levels, suggesting oxidative stress biomarkers could serve as early disease markers and aid in treatment decisions.

Oxidative stress in AFD involves an imbalance between ROS production during mitochondrial oxidative phosphorylation and the antioxidant system’s detoxification capacity [87]. Elevated plasma levels of 8-hydroxydeoxyguanosine (8-OHdG), a marker of oxidative DNA damage, are documented in AFD patients with cardiomyopathy [88]. This aligns with Simoncini et al.s’ findings of increased oxidative stress markers and decreased antioxidant defenses in AFD patients, even those untreated, showing early organ damage despite normal lyso-Gb3 levels [86].

Upregulation of iNOS and increased nitrotyrosine levels in AFD patients indicate oxidative and nitrosative stress. A deficiency in tetrahydrobiopterin (BH4), an essential NOS cofactor, was found in heart and kidney biopsies from AFD patients, linking BH4 deficiency to oxidative stress via reduced antioxidant capacity and NOS uncoupling [45]. This deficiency was not corrected by ERT, highlighting current treatments’ limitations.

Glutathione (GSH), a key antioxidant, was downregulated in male Fabry mouse models compared with females, suggesting sex-specific differences in AFD pathogenesis. Gb3 accumulation in AFD-iPSC-derived vascular endothelial cells suppressed superoxide dismutase 2 (SOD2) expression, leading to mitochondrial ROS production and vascular dysfunction [89]. Similar mitochondrial ROS increases and dysfunction were observed in AFD-iPSC-derived kidney organoids [90].

Biancini et al. revealed disturbances in GSH metabolism in AFD patients, increased lipid peroxidation, and elevated nitric oxide levels, correlating these changes with higher plasma levels of pro-inflammatory cytokines [91]. Chronic inflammation, including heparinase release, degrades the endothelial glycocalyx, explaining endothelial dysfunction in AFD [74]. Targeted treatments, such as protective Tie2 treatment, recombinant AGAL, heparin, anti-inflammatory and antioxidant drugs, and a specific angiopoietin-1 receptor (Tie2) inhibitor, have shown improvement in glycocalyx structure and endothelial function in vitro.

These findings highlight oxidative stress’s critical role in AFD pathogenesis, its contribution to early organ damage, and the potential benefits of oxidative stress-targeted therapies.

### 2.6. Endothelial Dysfunction 

The pathophysiology of AFD vascular pathology remains largely uncertain. Endothelial dysfunction in AFD has been documented, showing both altered FMD and serum biomarkers of endothelial dysfunction [58]. Potential mechanisms include the accumulation of Gb3 in the endothelium, SMCs proliferation, increased IMT, loss of vascular compliance, and endothelial dysfunction [62,92]. Ischemia significantly shapes the disease phenotype, affecting small vessels in the cerebrovascular system, heart, kidneys, peripheral nervous system, and skin, indicating systemic vasculopathy [11].

Vascular ischemic lesions in AFD result from vascular dysfunction involving endothelial dysfunction, cerebral perfusion alterations, and a pro-thrombotic shift [72,93]. Other cardiovascular risk factors may contribute to increased athero-thrombogenesis and arterial performance deterioration [94]. Gb3 accumulation triggers local renin-angiotensin system activation, upregulation of adhesion molecules, cytokines, chemokines, and pro-thrombotic factors, and reduced NO synthesis and bioavailability [62] (Figure 1).

Gb3 storage is thought to induce ROS production, Rho-kinase (ROCK) activation, and eNOS dysregulation [45,95,96], leading to muscular hypercontractility and vasospasm, potentially initiating AFD vasculopathy [92]. ROS-induced oxidative stress, overexpression of adhesion molecules, and cellular dysfunction are also significant factors [97]. ROS can cause irreversible damage to DNA, lipids, and proteins, contributing to the oxidation of low-density lipoproteins (LDL). ROS-induced transcription of cell adhesion molecules, mediated by NF-κB and other transcription factors, further contributes to Anderson–Fabry vasculopathy [45]. Specifically, E-selectin, ICAM-1, and VCAM-1 promote leukocyte rolling and adhesion at the endothelial level, initiating arterial wall infiltration and damage [28]. Immunohistochemical analyses by Moore et al. [98] revealed enhanced nitrotyrosine staining in dermal and cerebral blood vessels, indicating NO and ROS production dysregulation. eNOS uncoupling is detected in the presence of GLA deficiency, potentially altering vascular function. Clinical studies, such as Katsuta et al.s’ [99], associated hypertensive and arteriosclerotic vascular alterations with elevated serum adhesion molecules in Japanese AFD patients. Gb3 accumulation down-regulates KCa3.1 in cultured endothelial cells, inhibiting endothelium-derived relaxing factor (EDRF) production [58]. Lyso-Gb3, identified by Aerts et al. [45], induces SMCs proliferation in vitro and is more elevated in young male AFD patients compared with controls, suggesting early arterial medial layer involvement [62].

This leads to increased IMT, a more specific vascular wall alteration in AFD than traditional IMT changes seen in premature atherosclerotic disease [62]. Doran et al. [100] proposed that Gb3 accumulation and subsequent proliferation involve SMCs more than ECs, leading to intimal-media layer hypertrophy and inflammatory cell influx. This specific vasculopathy predominantly forms fibrotic structures rather than an enlarged atheroma. Hyperdynamic circulation combined with less elastic vascular walls may cause local renin-angiotensin system overregulation. Angiotensin II, through the AT1 receptor, initiates an inflammatory cascade reducing nicotinamide-adenine dinucleotide phosphate oxidase, ROS formation, and NF-κB expression, leading to increased adhesion molecule and chemokine expression and β-integrin upregulation, a key extracellular matrix modulator [101]. The role of the angiotensin 2 receptor (AT2) in AFD, particularly in early vascular damage as seen in diabetics, remains unexplored [62].

Storage of Gb3 within the arterial media layer promotes cell proliferation and fibrotic remodeling of the arterial wall, leading to increased stiffness and consequent shear stress. This may increase the expression of angiotensin 1 and 2 receptors in endothelial cells, ROS production, NF-κB, β-integrin, and cyclooxygenase 1 and 2 activity while decreasing nitric oxide synthesis [62]. These mechanisms contribute to coronary microvascular dysfunction (CMD), an important feature of Fabry cardiomyopathy. CMD has been described irrespective of LVH and gender, potentially representing the only sign of cardiac involvement, especially in females. The pathophysiological role of inflammation in CMD has been investigated in different clinical scenarios [102,103], highlighting the relation between the inflammatory profile and the presence and extent of CMD [104].

Lyso-Gb3 promotes the proliferation of SMCs and contributes to the thickening of the arterial intima-media layer in patients with AFD [62,80]. This stimulation leads to an influx of inflammatory cells to the arterial media layer, activation of the renin-angiotensin system, secretion of adhesion molecules and cytokines, and an increase in the ECM, determining a pro-inflammatory effect on leukocytes and endothelial cells. Increased ROS and decreased NO contribute to endothelial damage [62]. Various hypotheses suggest that endothelial Gb3 accumulation causes eNOS dysregulation, resulting in the formation of oxidant species derived from NO, which could be direct markers of vasculopathy in AFD [56,105,106,107,108]. Gb3 may also contribute to endothelial damage through other pathogenic mechanisms related to KCa3.1 channel dysfunction, innate immunity, altered autophagy, or mitochondrial function. Over time, this chronic inflammatory state can lead to the development of fibrosis in various tissues, primarily in the kidneys and heart, with consequent clinical manifestations [109]. 

Elevated levels of MPO in the blood are significant indicators of acute cardiovascular events, atherosclerosis, coronary stenosis, and endothelial dysfunction, furthering the formation of atherosclerotic plaques by promoting lipid peroxidation and negatively impacting left ventricular function. In their study, Kaneski et al. found that male patients with AFD exhibit significantly higher MPO levels compared with controls, while female heterozygotes also show elevated levels, though not statistically significant [75]. Elevated MPO levels correlate with a higher risk of vascular events over the years, and ERT failed to reduce MPO levels even after 55 months of treatment, highlighting a persistent issue. The exact mechanism behind elevated MPO in AFD is not entirely understood. MPO is produced by neutrophils, monocytes, and macrophages in response to ROS, suggesting that neutrophils are particularly activated in AFD, enhancing inflammatory interactions and leukocyte adhesion to endothelial cells. Gb3 also contributes to cell adhesion and B-cell apoptosis. The persistent elevation of MPO despite ERT suggests the need for new therapies to reduce vascular events in Fabry disease.

In summary, SMCs are primarily involved in AFD vasculopathy, with early stages characterized by angiotensin 2 overproduction, SMCs proliferation, and Gb3 storage causing intimal-media layer thickening. Early intervention, including enzyme therapy and renin-angiotensin system blockers, may be beneficial in preventing or delaying vascular complications in AFD [62].

## 3. Inflammation-Related Cardiovascular Complications: Clinical Correlates

### 3.1. Role of Inflammation in Myocardial Involvement

Several molecular pathways have been reported to contribute to the inflammatory activation in AFD, including the NF-κB pathway, oxidative stress, and the transforming growth factor-β (TGF-β) pathway. Inflammatory activation is closely related to autophagy, where defective autophagic proteins could enhance inflammasome activation and sphingolipid homeostasis. In vitro studies show that intra-lysosomal Gb3 impairs endocytosis and autophagy, induces apoptosis, and interferes with mitochondrial energy production [80].

Increased levels of lymphocytes and macrophage-related markers CD68, CD163, and CD45 in endomyocardial biopsy samples from AFD patients have been documented, supporting the concept of AFD as an “inflammatory cardiomyopathy” [110,111]. Cardiovascular magnetic resonance (CMR) imaging studies with T1 and T2 mapping assessing myocardial lipid content and inflammation at different disease stages suggest a central role for inflammation in early AFD progression [112,113]. Clinical and experimental evidence also supports the role of inflammation in AFD and other lysosomal storage disorders [77,85,114,115,116]. Deficiency of α-Gal A limits degradation, favoring the accumulation of lipidic antigens, while Gb3 and lyso-Gb3 may act as antigens themselves, activating invariant natural killer T-cells and leading to chronic inflammation and autoimmunity. Glycosphingolipid-mediated effects are abolished by anti-TLR4 antibodies, indicating the importance of this pathway in promoting TGF-β-mediated extracellular matrix remodeling and fibrosis [77,85,114,115,116]. Defective autophagy also promotes inflammation through inflammasome activators and the release of ROS [77]. Chronic inflammatory activation was observed in endomyocardial biopsies from patients with AFD. Knott et al. [117] recently linked myocardial inflammation with microvascular dysfunction and perfusion abnormalities in early cardiac involvement. 

A model proposed by Nordin et al. [118] suggests an evolution of myocardial phenotype in AFD consisting of an accumulation phase, a hypertrophy and inflammation phase, and a fibrosis and impairment (late) phase. The silent storage phase starts in childhood and is subclinical, with normal but falling myocardial T1 values. The overt storage phase indicates T1 is low, progressing faster in men than women, and is associated with left ventricular (LV) mass within normal limits and electrocardiogram (ECG) changes. The myocyte hypertrophy and inflammation phase involve late gadolinium enhancement (LGE) and inflammation primarily in the basal inferolateral wall, associated with persistent chronic troponin elevation but no thinning, potentially occurring before LVH in women. The fibrosis and impairment phase is characterized by persistent LVH, troponin elevation, fibrosis (myocyte death), thinning, extensive LGE, NT-proBNP (N-terminal pro-brain natriuretic peptide) elevation, LV impairment, and clinical heart failure. Nordin [112] showed that LGE in established AFD is chronic inflammation strongly correlating with troponin levels, supported by findings from positron emission tomography/magnetic resonance imaging (PET/CMR) studies [119] and endomyocardial biopsy [120].

Myocardial inflammation plays a key role in cardiac AFD, supported by frequent observations using CMR of high T2 mapping values, mainly in areas of LGE, typically representing edema, and increased troponin indicating myocardial injury. Di Taranto et al. [121] correlated histology with T2 and myocardial injury in AFD, finding that mild myocardial macrophage/lymphocyte infiltrates are common in AFD and associated with higher T2 values, but not with altered troponin levels. Persistent T2 and troponin elevation over one year suggested chronic myocardial edema and injury, with associated clinical deterioration. These findings could demonstrate an essential role for inflammation in AFD pathogenesis, with potential therapeutic implications [122].

The accumulation of Gb3 in the lysosomes of vascular cells, particularly in myocardiocytes in the cardiac phenotype, causes alterations in energy metabolism, mitochondrial dysfunction, and activation of inflammatory molecules, leading to autoimmune myocarditis [16]. The accumulation of Gb3 and its derivatives in cardiomyocytes induces an inflammatory response, increased oxidative stress, and apoptosis [123] (Figure 2). 

Several mediators involved in pathogenic mechanisms such as immune response activation, proliferation and hypertrophy of cardiomyocytes, oxidative stress, and fibrosis development are associated with Fabry cardiomyopathy [124,125]. Systemic vasculopathy and its derived inflammatory cascade might contribute to the progression of organ injury related to cardiomyopathy and nephropathy [52,80,124]; T1 values appear to be earlier markers than troponin, relating to lipid deposition [126]. Circulating levels of troponin increase in Fabry patients with LGE and are associated with increased T2 values [112]. 

### 3.2. Macrovascular Alterations in Fabry Disease

Studies have shown that AFD patients exhibit increased IMT in the carotid, brachial, and abdominal aorta and reduced FMD in the brachial artery compared with healthy subjects [55,56]. Boutouyrie et al. [54] observed accelerated arterial wall hypertrophy in medium-sized vessels in AFD patients, with radial artery IMT widening significantly more rapidly in AFD patients than in healthy individuals. Kalliokoski et al. [56] noted the most considerable differences in abdominal aorta IMT, aligning with McGill et al.s’ [127] findings that atherosclerotic lesions initially develop there.

Costanzo et al. [48] found that even AFD Mutation Carriers (AFDMCs) had increased carotid IMT without plaques and reduced brachial artery FMD. This macrovascular involvement in AFD includes rapid radial artery wall hypertrophy and common carotid artery (CCA) IMT [55]. The disproportionately small amount of Gb3 deposits compared with carotid artery wall mass suggests mechanisms beyond lipid storage (Figure 2).

General population studies associate increasing IMT with cardiovascular risk factors like aging, high LDL cholesterol, hypertension, smoking, type 2 diabetes, and microalbuminuria. Increased carotid IMT predicts coronary atherosclerosis, coronary disease, and stroke development, serving as a surrogate marker for generalized atherosclerosis [128]. Barbey et al.s’ study on hemizygous and heterozygous AFD patients compared with age-matched controls revealed that only age correlated with common carotid artery IMT [44]. Most AFD patients in Barbey et al.s’ study [129] had traditional cardiovascular risk factors, contributing to notable carotid wall thickening. Interestingly, despite increased IMT, no carotid bifurcation plaques were observed in these patients. Similar alterations in radial artery IMT, a medium-sized muscular artery, were also noted [55], a phenomenon unique to AFD among lysosomal storage disorders. Preserved or improved vasomotricity and vascular elasticity in AFD might represent adaptive changes providing apparent “cardioprotection” against the enzymatic defect.

Cross-sectional studies indicate a stronger carotid IMT relation with more cardiovascular risk factors [130,131], and increased carotid IMT correlates with higher risks of coronary disease and stroke even after adjusting for other cardiovascular risk factors [132,133]. 

AFD is associated with marked non-atherosclerotic arterial thickening. Evidence suggests increased IMT as a future cardiovascular event indicator, and preventive measures can reduce complication incidence. Whether carotid artery IMT stabilization or regression is a valid treatment efficacy indicator in AFD patients remains an area for further clinical research.

### 3.3. Microvascular Alterations in Fabry Disease

AFD patients accumulate endothelial Gb3 deposits in vessel walls of varying sizes, leading to multisystemic symptoms. Nailfold capillary changes, identified as early as 1993 in connective tissue diseases like systemic sclerosis and CREST, have been recognized as diagnostic in AFD. Wasik et al.s’ (2009) case-control study on 25 AFD patients revealed significant atypical capillaries, primarily bushy or clustered, compared with healthy controls, with other connective tissue disease characteristics like giant capillaries and avascular fields being absent [53]. 

Altered endothelial function due to glycosphingolipid accumulation and altered terminal organ perfusion from arteriovenous shunting and inadequate capillary perfusion may explain these NFC alterations [50,55,56,134]. This endothelial dysfunction could result in structural lesions at the shunt level and compromise sympathetic vasomotor control due to small-fiber neuropathy. Endothelial cells mediate smooth muscle activity through the release of vasodilatory prostaglandins and vasoconstrictors like endothelin-1. Sphingolipid accumulation in AFD may disrupt these regulatory mechanisms.

Nailfold capillary morphological changes in Fabry patients are similar in hemi- and heterozygous individuals and are not age-dependent. Despite renal failure being common in advanced AFD stages, no severe microangiopathy was noted in patients with significant nephropathy. Additionally, there is a high incidence of Raynaud’s phenomenon in Fabry patients, linked to pathological capillary changes observed in NFC (Figure 2). 

Costanzo et al.s’ observational study from 2008 to 2011 on AFDMCs showed capillaroscopic alterations, including irregular architecture, avascular fields, atypical capillaries, abnormal density, hemorrhages, and apical dilation, compared with healthy controls. AFDMCs exhibited twice the rate of irregular architecture and significantly different atypical capillaries [48].

Deshayes et al.s’ [57] cross-sectional study on French AFD patients noted a higher rate of Raynaud’s phenomenon and branched capillaries, especially in males, reflecting severe microcirculatory involvement. The study reported a 38% incidence of Raynaud’s phenomenon in Fabry patients compared with 5% in controls, with literature showing a 0.5–16% prevalence in the general male population and 2.5–22% in females. Chronic occupational exposure in men may explain the higher incidence of dystrophic capillaries in this group.

Glycosphingolipid accumulation induces small fiber neuropathy, endothelial dysfunction, and SMCs proliferation, affecting vascular tone and potentially triggering Raynaud’s phenomenon, which may cause extremity pain in nearly half of Fabry patients.

Summarizing, Nailfold Capillaroscopy is a valuable diagnostic tool for assessing microcirculatory function in AFD. Early anomalies detected by Nailfold Capillaroscopy warrant close follow-up with instrumental and laboratory tests, even without clinical damage. Further studies are needed to evaluate if early ERT improves survival and prognosis. The response to therapy heavily depends on the status of organ involvement, and future therapeutic strategies should focus on preserving functional organs in these patients.

### 3.4. Pathophysiological Differences between Fabry Disease and Atherosclerosis

AFD and atherosclerosis both involve endothelial dysfunction and inflammation, but their pathophysiological mechanisms differ significantly. AFD’s vascular damage is driven by the accumulation of Gb3, leading to oxidative stress, impaired NO synthesis, and a specific inflammatory response triggered by Gb3 accumulation. This accumulation disrupts eNOS function, causing eNOS uncoupling and the formation of superoxide instead of NO, leading to oxidative stress and vascular inflammation [77,82]. Additionally, the upregulation of adhesion molecules and the activation of the local renin-angiotensin system further exacerbate vascular damage and inflammation [77,82]. Unlike atherosclerosis, AFD rarely leads to obstructive coronary artery disease. Instead, patients exhibit increased IMT in carotid arteries and coronary microvascular dysfunction without significant plaque formation, leading to non-obstructive myocardial infarctions [29,135].

In contrast, atherosclerosis is driven by lipid accumulation and immune responses within the arterial walls. The process begins with the deposition of oxidized LDL in the arterial intima, leading to endothelial activation and the recruitment of monocytes. These monocytes differentiate into macrophages, engulf lipids, and form foam cells, initiating plaque formation [136,137]. Persistent inflammation involves various immune cells, including T cells, dendritic cells, and macrophages, which secrete pro-inflammatory cytokines (e.g., TNF-α, IL-6) and contribute to plaque growth and instability [138]. Atherosclerotic plaques can rupture, leading to thrombus formation and acute cardiovascular events such as myocardial infarction and stroke [139]. Atherosclerosis is strongly associated with traditional cardiovascular risk factors, including hypercholesterolemia, hypertension, diabetes, smoking, and elevated levels of lipoprotein(a) and uric acid, which exacerbate endothelial dysfunction and promote plaque development [136,140].

In terms of clinical presentation, AFD typically does not feature flow-limiting plaques but rather diffuse IMT and microvascular dysfunction. AFD patients exhibit increased IMT in carotid and coronary arteries without significant atheromatous plaques, leading to myocardial infarctions not caused by occlusive coronary artery disease but rather by microvascular dysfunction and endothelial impairment. This is evident in studies demonstrating that Fabry patients have higher IMT but no significant coronary artery stenosis, with myocardial infarctions being type 2 rather than type 1 [82]. To synthesize the main differences, we can focus on the following points: -Molecular Pathways: In AFD, the primary driver is the accumulation of Gb3, leading to oxidative stress and inflammation, whereas in atherosclerosis, lipid accumulation and subsequent chronic inflammation are central.-Immune Cells and Cytokines: AFD involves a specific inflammatory response triggered by Gb3 accumulation, whereas atherosclerosis involves a broader range of immune cells and cytokines associated with lipid-induced inflammation.-Clinical Manifestations: AFD typically does not result in obstructive coronary artery disease but is associated with microvascular dysfunction and non-obstructive myocardial infarctions. Atherosclerosis, in contrast, leads to plaque formation and obstructive cardiovascular events.

Understanding these differences is important for developing targeted therapeutic strategies. In AFD, treatments focus on reducing Gb3 accumulation and addressing its metabolic consequences, whereas in atherosclerosis, therapies aim to manage lipid levels, control inflammation, and stabilize plaques.

### 3.5. Role of Fibrosis in Fabry Disease

Both TNFR1 and TNFR2 (Tumor Necrosis Factor Receptor 1 and 2) levels are elevated in AFD and associated with LGE at CMR, an early indicator of cardiac hypertrophy [83]. These receptors are potential therapeutic targets for heart failure management [141]. Increased BNP and MR-proANP (Midregional pro Atrial Natriuretic Peptide) levels in AFD patients with LGE and diastolic dysfunction suggest significant long-term pathological cardiac remodeling. Matrix metalloproteinases MMP2 and MMP9 play crucial roles, with MMP2 linked to HFpEF and MMP9 to LVH and adverse ECM remodeling [129]. Elevated galectin-1 levels support ECM remodeling’s significant role in AFD, while galectin-3 is a biomarker for advanced disease and heart failure progression [142].

Studies show a correlation between elevated serum levels of collagen metabolism biomarkers and left ventricular mass and malignant ventricular arrhythmias [143,144] in AFD patients. Increased serum procollagen type I carboxyterminal pro-peptide (PICP), a marker of collagen synthesis, has been observed in early AFD stages, even without cardiac dysfunction. Suppression of MMPs, essential for collagen degradation, may exacerbate myocardial collagen deposition [144]. Monitoring these biomarkers provides valuable insights into the disease’s progression, aiding in assessing AFD’s cardiac impact.

ERT effectiveness in AFD cardiac involvement is influenced by fibrotic response severity. This association underscores the clinical implications of myocardial fibrosis in AFD, highlighting the necessity for early detection and targeted therapeutic strategies. TGF-β1, a pro-fibrotic cytokine, is central to fibrosis, involving fibroblast activation and myofibroblast trans-differentiation, leading to ECM deposition [145]. Elevated plasma TGF-β1 levels are observed consistently in lysosomal storage disorders, including AFD. Lyso-Gb3 can activate vascular ECs, resulting in TGF-β1 secretion and amplifying fibrosis. However, Choi et al. [146] found that lyso-Gb3 could inhibit proliferation, differentiation, and collagen synthesis in mouse adventitial fibroblasts through downregulation of the Ca^2+^-activated potassium channel (KCa3.1). Elevated serum renin levels, correlated with plasma lyso-Gb3 levels in AFD patients, lead to increased Angiotensin II production, stimulating cardiac fibroblast activation, collagen deposition, Alpha Smooth Muscle Actin (α-SMA) production, and TGF-β1 secretion [147,148].

Cardiac fibrosis has a central role in AFD progression, driven by pro-inflammatory and pro-fibrotic cytokines through pathways such as TGF-β, NF-κB, MAPK/ERK (Mitogen-activated protein kinase/Extracellular signal-regulated kinase), and the renin-angiotensin system. The presence and severity of fibrosis are critical determinants of the therapeutic response to ERT, marking a critical juncture in AFD beyond which current treatments become less effective. This underscores the need for innovative therapeutic approaches to manage cardiac fibrosis in AFD.

## 4. Established and Emerging Inflammatory Biomarkers Related to Fabry Disease Burden and Progression and New Frontiers in Treatment

Recent advances in understanding AFD have underscored inflammation’s critical role in its pathogenesis and the significance of novel biomarkers for tracking disease progression (Table 2). These discoveries offer insights into AFD’s mechanisms and open avenues for improved diagnosis and treatment.

A promoter polymorphism (−174G > C) of the IL-6 gene is linked to the Mainz Severity Score Index (MSSI) in AFD patients, with the IL-6 C allele influencing MSSI, especially in females [149]. Chronic inflammation in AFD is driven by glycolipids like Gb3 and lyso-Gb3, which interact with TLRs, triggering inflammation and fibrosis in organs such as the kidneys and heart [77]. Cultured PBMC from AFD patients show heightened pro-inflammatory cytokine production, with Gb3 likely mediating this via TLR4. Invariant natural killer T (iNKT) cells exhibit a pro-inflammatory phenotype in AFD [73,150].

Proteomic analyses have identified up-regulated proteins such as γ-enolase and galectin-1 in AFD patients, highlighting their potential as biomarkers [151]. Elevated serum C-reactive protein levels before stroke onset link persistent inflammation to recurrent strokes in AFD patients [152]. Elevated levels of inflammatory markers like TNF, TNFR2, IL-6, and MMP-2 in AFD patients with left ventricular hypertrophy and increased BNP, MR-proANP, MMP-2, and galectin levels in renal dysfunction, correlate with disease progression [116].

Complement activation products C3a and C5a are increased in AFD patients with anti-drug antibodies, and elevated IL-6, IL-10, and TGF-β1 levels suggest complement activation’s role in renal damage [153]. Traditional biomarkers like serum creatinine and proteinuria are limited in monitoring AFD, while elevated serum GDF-15 (Growth/differentiation factor 15) and syndecan-1 levels in patients with cardiomyopathy and nephropathy indicate potential associations with fibrosis and clinical outcomes, making them helpful biomarkers to identify classic AFD patients with cardiac and renal involvement. Elevated syndecan-1 levels are linked to conditions such as ischemic heart disease, acute heart failure, lupus nephritis, and chronic and acute kidney injury. Additionally, GDF-15 is induced in response to injury and is associated with cardiovascular damage and CKD. GDF-15 also plays a protective role against tissue injury by inhibiting myocardial hypertrophy and fibrosis [154]. Myocardial inflammation of autoimmune origin in AFD cardiomyopathy, with circulating anti-Gb3 antibodies as biomarkers, has been identified [155].

Elevated serum IL-6 and TNF-α levels in AFD patients treated with ERT correlate with MSSI scores, indicating a greater disease burden [156]. Both AFD and hypertension activate the innate immune system, with TLR4 as a common trigger, and the renin-angiotensin system contributes to chronic inflammation and oxidative imbalance, suggesting potential targeted therapies [97].

Gb3 accumulation causes lysosomal dysfunction, affecting the lysosome-autophagy-mitochondria interaction, which is crucial for organ damage in AFD [157]. Gb3 deposition in cardiomyocytes increases cardiac excitability by altering calcium activity [158,159]. The mTOR (mammalian Target Of Rapamycin) pathway is critical for regulating mitochondrial metabolism and autophagy [160,161]. Dysregulation of this pathway inhibits mitochondrial metabolism in AFD cells.

Biomarkers such as symmetric dimethylarginine (SDMA), l-arginine, asymmetric dimethylarginine (ADMA), and l-homoarginine (hArg) reflect extracellular matrix turnover and endothelial dysfunction in AFD patients [72]. Lyso-Gb3 inhibits NO synthase, contributing to vasculopathy and cardiac symptoms, which are mitigated by paricalcitol or calcitriol [62,162]. Complement pathway interactions and elevated microRNAs (miRNAs) suggest their role as biomarkers for AFD [163].

Incorporating these markers into clinical practice could enhance diagnostic accuracy, especially in asymptomatic or early-stage patients [109]. Elevated levels of TGF-β1, α-TGF, FGF2, and VEGF-A in AFD correlate with myocardial fibrosis and adverse cardiovascular events, serving as prognostic indicators [29]. Elevated 3-nitrotyrosine (3-NT) levels may serve as biomarkers for vascular involvement and treatment response [109].

Emerging serum biomarkers like MMP-9, angiostatin, SDMA(symmetric dimethylarginine), and the hArg/SDMA ratio indicate extracellular matrix turnover and endothelial dysfunction [82]. These findings highlight inflammation’s significant role in AFD pathogenesis and the potential of novel biomarkers to enhance disease monitoring and therapeutic strategies. By targeting inflammatory pathways, outcomes for AFD patients may improve. Monitoring cardiac remodeling and systemic inflammation markers could provide increased sensitivity for early detection, aiding the timely initiation of ERT to prevent complications [164].

## 5. A Look at Future Perspectives in Therapy, Biomarkers, Molecular Targets, and Mechanisms

The detailed mechanisms and consequences of inflammatory responses in AFD remain incompletely understood. The therapeutic strategy targeting inflammation is highly demanding and appealing [165]. 

AFD management remains challenging despite advancements in ERT. While ERT has demonstrated efficacy in improving organ function, it requires biweekly intravenous administration, which poses risks such as immunogenicity that can ultimately reduce its effectiveness [166]. Anti-drug antibodies often develop within three to six months after starting ERT, predominantly affecting males with classic AFD. These antibodies can bind to the therapeutic enzyme, leading to its inactivation and reduced cellular uptake, thus diminishing ERT efficacy and worsening clinical outcomes [166,167]. Strategies to mitigate these effects, such as increasing ERT dosage and using immunosuppressive therapies, have shown varied long-term effects and potential side effects [167] (NCT03614234). 

Despite these challenges, significant progress is being made in developing new therapeutic approaches for Fabry disease. Ongoing research and clinical trials are focused on more effective and less burdensome treatments, including oral therapies, novel ERT formulations, gene therapy, and substrate reduction therapy (SRT). 

One promising oral therapy is migalastat, a pharmacological chaperone that stabilizes the α-GalA enzyme, facilitating its proper trafficking and function. Clinical trials such as FACETS and ATTRACT [37,168] have demonstrated its efficacy in reducing left ventricular mass index and stabilizing eGFR. Substrate reduction therapies are another exciting area of research. Drugs like lucerastat and venglustat inhibit glycosphingolipid synthesis, aiming to reduce Gb3 accumulation in tissues. These therapies are being tested for their potential to modify the disease course and improve symptoms, offering a new avenue for patients with non-classical mutations or those who could benefit from a combined approach with ERT (MODIFY trial, NCT03425539; NCT0528054; NCT05206773).

Lucerastat, an orally active glucosylceramide synthase (GCS) inhibitor, aims to restore enzyme activity without intravenous administration by blocking the synthesis of a precursor molecule of Gb3. Phase 1 trial results (NCT02930655) showed lucerastat was well tolerated and significantly reduced blood levels of Gb3 and related fatty molecules compared with ERT alone, though no effects were observed on kidney or heart function. Most participants opted to continue in an open-label extension study (NCT03737214), where interim results for up to two years showed maintained reductions in Gb3. While kidney function had been declining in participants before entering MODIFY, data from the Phase 3 trial and its extension study showed lucerastat associated with slower declines compared with a placebo and historical data. Improvements in cardiac health measures were also observed, including a decrease in the left ventricular mass index, a predictor of cardiovascular disease.

Venglustat, another SRT, is being evaluated in the CARAT trial (NCT05280548) for its effect versus standard therapy on left ventricular mass at CMR and organ damage in Fabry disease patients with LVH. 

AL01211, an investigational therapy, has shown promise in Phase 1 trials by safely lowering levels of Gb3 in healthy adults. This oral treatment is currently in a Phase 2 trial (NCT06114329) involving men with classic Fabry disease who have never used a disease-related treatment. AL01211 aims to maximize its effects on targeted organs and minimize side effects on the brain by avoiding the central nervous system.

In addition to oral therapies, new formulations of ERT are being explored. Pegunigalsidase alfa, a PEGylated ERT, has shown sustained plasma concentrations and efficacy in reducing plasma lyso-Gb3 and its deposition in the kidney (slowing eGFR slope and proteinuria) and in the left ventricle, potentially offering a less immunogenic alternative to current ERT options, as PEGylation could mask some epitopes from the immune system, reverse anti-drug antibodies, and induce immune tolerance [169,170,171,172]. Ongoing clinical trials (NCT03614234, NCT06095713, NCT05710692, and NCT03566017) will evaluate its efficacy and safety in the real world compared with previously approved ERTs. Another innovative approach involves a plant-based ERT derived from Phycomitrella patens, designed to elude immunogenic responses, that has been shown to reduce urinary Gb3 levels [173,174].

Gene therapy holds significant promise for Fabry disease, with several approaches under investigation. One method uses adeno-associated virus (AAV) vectors for one-time intravenous administration to deliver the GLA gene to hepatocytes, enabling endogenous enzyme production and secretion. Clinical trials (NCT06270316, NCT04519749, STAAR—NCT04046224) are exploring the safety and efficacy of these therapies. Another approach involves lentivirus-mediated gene therapy, using lentivirus-transduced hematopoietic stem/progenitor cells to express α-galactosidase A. Early results are promising, showing increased enzyme activity and reduced Gb3 and lyso-Gb3 levels (NCT02800070). Additionally, mRNA therapy, encapsulating GLA mRNA in lipid nanoparticles, has shown potential in preclinical studies to increase enzyme levels in the liver, heart, and kidneys.

Moreover, combination therapies and novel administration protocols are being investigated to enhance the effectiveness of current treatments. The SHORTEN study (NCT06019728) aims to optimize ERT infusion protocols to reduce treatment burden by increasing the infusion rate and reducing infusion volume. The RECAFTURE trial (NCT02469181) evaluates the impact of ERT on cardiac function using advanced imaging techniques to understand its effects on left ventricular diastolic function and flow.

Other ongoing trials in Fabry disease focus on the role of inflammation in diagnosis and its correlation with imaging. The study NCT06226987 combines PET and CMR to identify myocardial inflammation, distinguishing between edema and fibrosis. The Bio-FAIR trial (NCT06007768) examines immune response biomarkers and their relationship with disease severity, quality of life, and pain in patients. The study NCT05698901 aims to develop an algorithm using biomarkers to evaluate disease progression and treatment response. Lastly, the FASHION study (NCT05761834) explores differences in inflammatory profiles between Fabry disease and hypertrophic cardiomyopathy, analyzing correlations between inflammatory phenotypes and cardiac severity through proteomic and transcriptomic analyses, echocardiography, and electrocardiography.

The future of Fabry disease treatment looks promising, with these innovative approaches potentially improving patient outcomes and quality of life.

Future research should focus on more comprehensive and effective management of lysosomal storage disorders, including AFD, by addressing the interplay of metabolic dysfunction, oxidative stress, and inflammation inherent to the disease [175]. This includes targeting mitochondrial functions and integrating antioxidant therapies with existing treatments. For example, coenzyme Q10 treatment, involved in ROS scavenging and mitochondrial ATP production, has been reported to alleviate mitochondrial dysfunction and reduce ROS production in Gaucher macrophages and NP-C patient-derived fibroblasts [176]. Antioxidants such as N-acetylcysteine are known to normalize pro-inflammatory cytokine production in Niemann-Pick disease type C fibroblasts, potentially through restoring trafficking and mitigating oxidative stress [177]. The antioxidant Glutathione has demonstrated effectiveness in attenuating oxidative stress in renal AFD in vitro models [90], while ascorbate supplementation has been found to decrease cerebral hyperperfusion in AFD patients undergoing ERT. The effect of ERT can be enhanced with adjunct antioxidants like vitamin E and ticlopidine, suggesting a multi-pronged approach may benefit AFD patients [178,179].

Exosomes represent a novel approach, both as biomarkers and therapeutic vehicles. They are signaling vehicles that mediate inflammatory processes and deliver therapeutic proteins effectively. Exosomes loaded with cytokines become biologically active upon interaction with target cells, such as macrophages, revealing a pro-inflammatory action [180]. In vitro and in vivo studies showed that Extracellular Vesicles carrying GLA (EV-GLA) were rapidly taken up and driven directly to lysosomes, restoring enzyme functionality more efficiently than clinical-reference agalsidase-alfa [181]. This represents a crucial advantage over ERT, which fails to access the brain [182].

Personalized medicine, integrating these advanced therapies tailored to individual disease profiles, represents a promising direction for the comprehensive management of AFD and other lysosomal storage disorders. This approach aims to address the interplay of metabolic dysfunction, oxidative stress, and inflammation inherent to the disease, potentially revolutionizing patient care [165,175].

## 6. Conclusions

AFD is a complex lysosomal storage disorder characterized by glycosphingolipid accumulation that leads to multi-organ damage. This review underscores the critical roles of oxidative stress, endothelial dysfunction, and inflammation in the disease’s pathogenesis. Elevated levels of advanced oxidation products and diminished antioxidant defenses contribute significantly to vascular and organ damage. Endothelial dysfunction arises from the intricate interplay of inflammatory pathways, including the TGF-β, NF-κB, and renin-angiotensin systems, which promote fibrosis and impair vascular function.

Understanding the inflammatory pathways and mechanisms in AFD, along with the identification of novel inflammatory biomarkers and the application of advanced imaging techniques, offers new insights for developing personalized therapeutic approaches. These advancements suggest potential strategies that extend beyond ERT, providing opportunities for more effective early interventions in the disease’s pathophysiology. The incorporation of biomarkers such as γ-enolase, galectin-1, and microRNAs, combined with cutting-edge imaging modalities, facilitates early diagnosis and precise monitoring of disease progression.

Emerging therapies show promise in stabilizing disease progression, while gene therapy and substrate reduction therapy represent potential future directions for long-term management. Targeting mitochondrial dysfunction and integrating antioxidant therapies offer promising avenues for mitigating oxidative damage and improving patient outcomes.

A deeper knowledge of AFD’s inflammatory mechanisms represents an exciting and evolving field. This comprehensive approach, leveraging early intervention and personalized medicine, holds the potential to significantly advance the effectiveness of treatments and improve clinical outcomes for patients with AFD. Further research in this domain is essential to fully elucidate these pathways and translate these insights into therapeutic innovations.

## Figures and Tables

**Figure 1 ijms-25-08273-f001:**
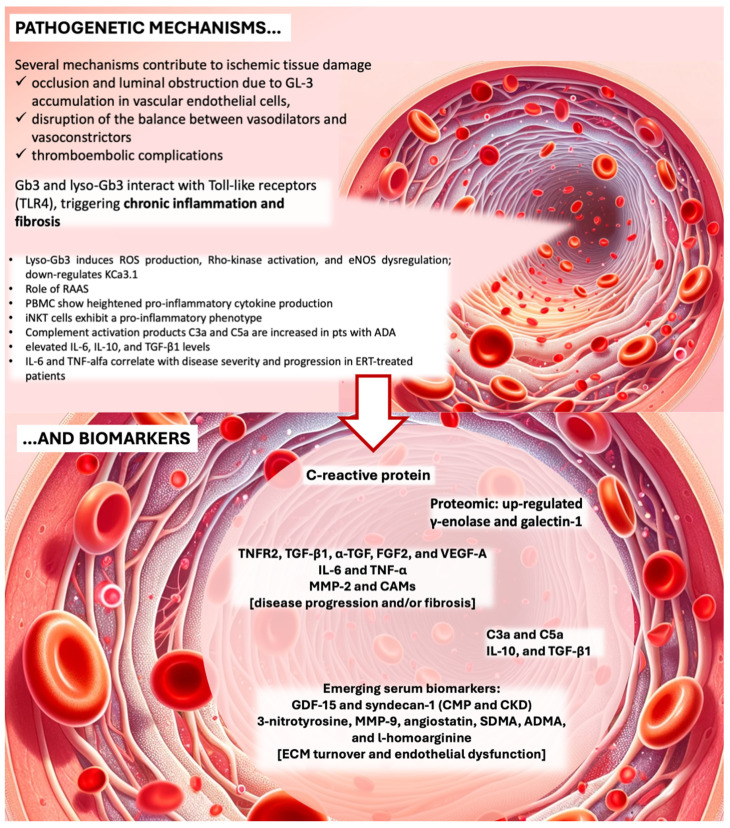
Endothelial disfunction, pathogenetic bases, mechanisms of endothelial dysfunction and biomarkers. Lyso-Gb3, lyso-globotriaosylceramide; ROS, reactive oxygen species; eNOS, endothelial nitric oxide synthase; KCa3.1, calcium-activated potassium channel 3.1; RAAS, renin-angiotensin-aldosterone system; PBMC, peripheral blood mononuclear cells; iNKT, Invariant natural killer T cells; ADA, antidrug antibodies; CMP, cardiomyopathy; CKD, chronic kidney disease; ECM, extracellular matrix; TNF, tumor necrosis factor; TNFR2, tumor necrosis factor receptor 2; IL, interleukin; TGF, transforming growth factor; FGF, fibroblast growth factor; VEFG-A, vascular endothelial growth factor; MMP, matrix metalloproteinase; CAMs, cell adhesion molecules; GDF-15, growth differentiation factor 15; SDMA, symmetric dimethylarginine; ADMA, asymmetric dimethylarginine. Credits: Some parts of the image were generated with the help of an artificial intelligence algorithm and then manually combined and annotated.

**Figure 2 ijms-25-08273-f002:**
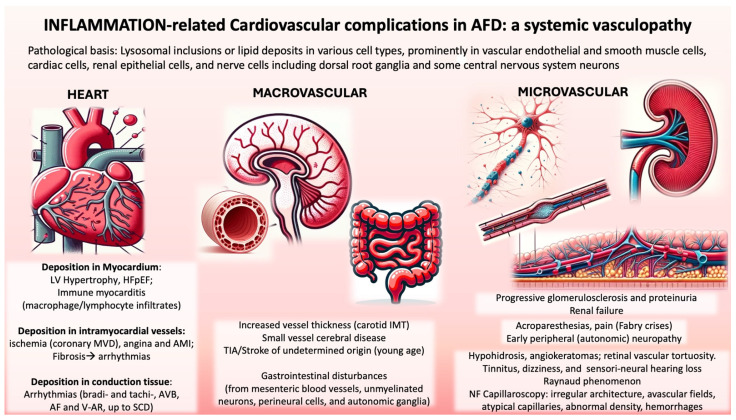
Overview of Anderson-Fabry Disease Complications related to cardiac and vascular inflammation are classified according to the mechanism (heart, macrovascular, and microvascular). Abbreviations: LV, left ventricular; HFpEF, heart failure with preserved ejection fraction; MVD, microvascular dysfunction; AMI, acute myocardial infarction; AVB, atrioventricular block; AF, atrial fibrillation; V-AR, ventricular arrhythmias; SCD, sudden cardiac death; IMT, intimal-media thickness; NF, nailfold. Credits: Some parts of the image were generated with the help of an artificial intelligence algorithm and then manually combined and annotated.

**Table 1 ijms-25-08273-t001:** Studies assessing macro- and microvascular complications in AFD patients. Abbreviations: CCA, common carotid artery; IMT, intima-media thickness; PWV, pulse wave velocity; FMD, flow-mediated dilatation; DUS, duplex ultrasound.

Authors	Study Design	Cases/Controls	Age (yo)	Sex	Type of Vessels	Outcome(s)	Results	Methods Employed
[52]	Case-control observa-tional	67/55	38.4 ± 14.3 M, 45.7 ± 13.3 F	27 M–40 F vs. 20 M–35 F	CCA, femoral arteries	Increase in IMT and PWM, reduction of FMD	IMT: +9% M, +8% F; PWV: +7% M, +4% F; FMD: −30% M, −5%	DUS (B-mode DICOM)
[53]	Cohort observa-tional	25	37.1 M, 41.8 F	17 M–8 F	Nailfold capillaries	Microangiopathy, functional and structural	Thick capillaries (62% vs. 10%), and other pathological patterns.	capillaroscopy (fluorescence videomicroscopy)
[48]	Case-control observa-tional	19/19	30.1 ± 14.8	3 M–16 F vs. 6 M–13 F	CCA, nailfold capillaries	Increase of IMT in CCA, FMD reduction, capillary alterations	IMT: +23% FMD: −32%; significant microangiopathy in nailfold capillaries in some cases	DUS (GE Vivid E) capillaroscopy
[54]	Case-control observa-tional	21/21	31 ± 13	21 M vs. 21 M	Radial artery	Increase of IMT in radial artery	+2.3 times more	DUS (high precision NIUS 02)
[55]	Case-control observa-tional	21/24	32 ± 13 M	21 M vs. 24 M	CCA, radial artery	Increase IMT in radial artery and CCA	CCA: +18% Radial art.: +2.3 times more	ecotracking systems with high definition (not reported)
[44]	Case-control observa-tional	53/120	45.0 ± 1.7 M 55.0 ± 2.2 F	24 M–29 F vs. 83 M–37 F	CCA	Increase of IMT in CCA, no plaques	+13% M +18% F	DUS (not reported)
[56]	Case-control observa-tional	17/34	38 ± 14	7 M–10 F vs. 16 M–18 F	CCA, brachial artery, and aorta	Increase of IMT, FMD reduction in brachial artery and aorta	IMT CCA: +11% IMT aorta: +27%; IMT brachial: +16%; FMD: −33%	DUS (Acuson)
[57]	Trasversal observa-tional	32/39	45.5 ± 13.8 48.2 ± 11.5	10 M–22 F vs. 24 M–15 F	Nailfold capillaries	Prevalence of Raynaud in AFD patients	Raynaud +38% than controls (5%)	Capillaroscopy (CapXview HD, Xport technologies, Craponne, France)

**Table 2 ijms-25-08273-t002:** Inflammatory biomarkers/mediators involved in the pathophysiological mechanisms of vascular damage in Fabry disease.

Mediators	Role in Vascular Damage/Fibrosis	Clinical Implications	Mechanism of Action	Inflammatory Cells Involved
Gb3/LysoGb3	Accumulation in endothelial cells, disrupts eNOS, reduces NO production, increases ROS production	Endothelial dysfunction, oxidative stress, chronic inflammation, multi-organ involvement	Direct accumulation in cells; potent inflammatory mediator	ECs, SMCs
eNOS	Uncoupling due to Gb3 accumulation, produces superoxide instead of NO; reduced NO bioavailability due to Gb3 accumulation and eNOS uncoupling	Oxidative stress, endothelial damage, increased ROS production → impaired vasodilation, increased risk of thrombosis	Enzyme dysfunction	ECs
ROS	Increased production due to eNOS uncoupling and RAS activation, causes oxidative stress	Oxidative stress, chronic inflammation, tissue damage	Oxidative damage	ECs, SMCs, neutrophils
ICAM-1, VCAM-1	Promote leukocyte adhesion and infiltration, driving chronic inflammation	Vascular inflammation, progression of endothelial damage	Increased expression	ECs, leukocytes
TNF-α, IL6	Chronic inflammation, endothelial dysfunction	Increased cardiovascular risk, disease progression	Cytokine signaling	Macrophages, T cells, ECs, SMCs
Mitochondrial Dysfunction	Secondary to lysosomal dysfunction, affects metabolic homeostasis	Altered metabolism and energy, cellular dysfunction, cell death	Disrupted metabolic pathways	ECs, SMCs
CRP	Nonspecific marker of chronic low-grade systemic inflammation, induces cytokines production (IL6, TNF-alfa, IL1), reduces eNOS activity and NO-mediated vasodilation	Indicates ongoing vascular injury and inflammation, prognostic value for CV events	Inflammatory biomarker	Macrophages, hepatocytes, ECs, SMCs, lymphocytes
VEGF	Promotes angiogenesis and endothelial dysfunction	Associated with disease severity and endothelial dysfunction	Specific endothelial cell mitogen	ECs
MPO	Elevated in response to oxidative stress, contributes to vascular inflammation	Associated with oxidative stress, vascular inflammation, and endothelial dysfunction, predictor of acute CV events, accelerated atherosclerosis and coronary stenosis,	oxidation of LDLreduction in NO bioavailability leading to endothelial dysfunction, activation of MMPs	Neutrophils
C3a and C5a	Increased in response to inflammation, contribute to endothelial damage and chronic inflammation	Associated with renal damage and inflammation	influence lymphocyte activity, promoting proliferation and differentiation, recruitment and activation of dendritic cells, cross-talk with TLR, proinflammatory cytokine induction stimulating profibrotic pathways	Macrophages, neutrophils
Syndecan-1	Elevated in endothelial damage, reflects glycocalyx degradation	Indicator of vascular damage, correlates with disease severity and cardiac and renal involvement (heart failure and fibrosis)	Glycocalyx component	ECs
TGF-β1	Promotes fibrosis	Associated with renal and cardiac damage, LVH, fibrosis and disease progression	Profibrotic cytokine/growth factor: promotes fibrosis in response to chronic inflammation by enhancing the synthesis of ECM	Macrophages, FBs, ECs
FGF2	Elevated levels promote fibrosis and chronic inflammation, particularly in cardiac tissue	Associated with myocardial fibrosis and adverse CV events	Cytokine signaling (regulates angiogenesis, cell growth, and tissue repair)	FBs, ECs
miRNAs	Elevated levels associated with endothelial dysfunction and inflammation	Potential biomarkers for disease monitoring	Post-transcriptional regulation	ECs
IL-10	Anti-inflammatory cytokine,	Modulates inflammatory response, potential marker of disease activity	Cytokine signaling; elevated levels suggest an attempt to counteract inflammation	Macrophages, T cells, ECs
GDF-15	Elevated in response to inflammation and oxidative stress	Indicator of disease severity and progression (kidney injury and cardiovascular involvement and outcomes)	Stress response signaling; modulates renal and cardiac injury, possibly providing protection from tissue injury and fibrosis	Macrophages, ECs
MMP-2 and MMP-9	Indicate extracellular matrix remodeling	Associated with fibrosis and vascular damage (also cardiac and renal)	Extracellular matrix degradation	Macrophages, FBs

Abbreviations: ECs, Endothelial cells; SMCs, smooth muscle cells Gb3, Globotriaosylceramide; NO, Nitric Oxide; eNOS, Endothelial NOS; ROS, Reactive Oxygen Species; ICAM-1, Intercellular Adhesion Molecule 1, VCAM-1 (Vascular Cell Adhesion Molecule 1; TNF-α, Tumor Necrosis Factor Alpha; IL-6, Interleukin 6; CRP, C-Reactive Protein; VEGF, Vascular Endothelial Growth Factor; MPO, Myeloperoxidase; TLR, toll-like receptors; FBs, fibroblasts; TGF-β1, Transforming Growth Factor Beta 1; LVH, left ventricular hypertrophy; ECM, extracellular matrix; miRNAs, MicroRNAs; FGF2, Fibroblast Growth Factor 2; IL-10, Interleukin 10; GDF-15, Growth Differentiation Factor 15; MMP-2 and MMP-9, Matrix Metalloproteinases.

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
