# Peer review of "Inflammation, Oxidative Stress, and Endothelial Dysfunction in the Pathogenesis of Vascular Damage: Unraveling Novel Cardiovascular Risk Factors in Fabry Disease"

_ijms, 2024, doi:10.3390/ijms25158273_

Round 1

Reviewer 1 Report

Comments and Suggestions for Authors

In the present review Faro and colleagues provide in-depth information on  the genetic disorder "Anderson-Fabry disease, AFD" linked vascular disease and the underlying mechanisms. The authors have highlighted the role of mutation in alpha-galactosidase-A gene which disrupts lysosomal function and causing vascular complication by increasing globotriaosylceramide (Gb3). The review has covered most clinically relevant aspects of AFD associated vascular pathology. The review is well written and informative and easy to follow. I have a few suggestions that may further help improve the review.

1: The authors provide ample information on the cellular and biochemical evidence of vascular complications associated with AFD. They emphasize the role of both endothelial and smooth muscle cells. It would be nice if the authors can include a paragraph on in vitro experimental evidence(s) highlighting studies that show the deleterious actions of Gb3 on endothelial health (free radical, NO and COX 2 etc.) and on SMC growth (with potential underlying mechanisms).

2. Although the authors discuss the severity of the disease in women a small statement on Gender based differences and role of sex hormones in influencing AFD severity would add to the review.

Author Response

Response to Reviewer #1: Re: manuscript ijms-3063569 entitled “Inflammation, Oxidative Stress, and Endothelial Dysfunction in the Pathogenesis of Vascular Damage: Unraveling Novel Cardiovascular Risk Factors in Fabry Disease

 Dear Reviewer 1,

We thank you for your thorough review of our manuscript. We have revised our manuscript according to your helpful suggestions. Each comment is reproduced below in bold followed by our response. A tracked version of the revision with the changes highlighted has been uploaded.

In the present review Faro and colleagues provide in-depth information on  the genetic disorder "Anderson-Fabry disease, AFD" linked vascular disease and the underlying mechanisms. The authors have highlighted the role of mutation in alpha-galactosidase-A gene which disrupts lysosomal function and causing vascular complication by increasing globotriaosylceramide (Gb3). The review has covered most clinically relevant aspects of AFD associated vascular pathology. The review is well written and informative and easy to follow.

We thank the reviewer for the appreciative comments.

I have a few suggestions that may further help improve the review.

1: The authors provide ample information on the cellular and biochemical evidence of vascular complications associated with AFD. They emphasize the role of both endothelial and smooth muscle cells. It would be nice if the authors can include a paragraph on in vitro experimental evidence(s) highlighting studies that show the deleterious actions of Gb3 on endothelial health (free radical, NO and COX 2 etc.) and on SMC growth (with potential underlying mechanisms).

Thank you. We added a dedicated paragraph as suggested.

“Effects of Gb3 on Endothelial Health and SMC Growth in Fabry Disease”

In vitro studies consistently show that Gb3 exerts harmful effects on endothelial and smooth muscle cells (SMCs), contributing to vascular complications in Fabry disease. The accumulation of Gb3 in endothelial cells markedly increases the production of reactive oxygen species (ROS), impairing nitric oxide (NO) synthesis. This oxidative stress disrupts NO signaling and inhibits endothelial NO synthase (eNOS) activity, resulting in reduced NO bioavailability and endothelial dysfunction{Satoh, 2014 #186} {Shen, 2008 #121}. Furthermore, evidence suggests that the widespread intracellular accumulation of Gb3 disrupts other cellular components regulating endothelial function {Altarescu, 2001 #257}.Other studies reveal that total eNOS and phosphorylated eNOS, both membrane-associated, are downregulated in Gb3-loaded human endothelial cells (ECs), while iNOS expression is upregulated. This dysregulation of the L-arginine/NO pathways may adversely affect vascular function and blood flow to vital organs, contributing to the risk of vasculopathy and cardiovascular complications {Namdar, 2012 #241}. Additionally, Gb3 has been shown to upregulate the expression of cyclooxygenase-2 (COX-2), a pro-inflammatory enzyme that exacerbates endothelial inflammation. This upregulation is linked to enhanced ROS production, creating a feedback loop that further damages endothelial function. COX-2-derived prostaglandins may play a compensatory role for decreased NO bioavailability, potentially explaining some cardiovascular effects associated with COX-2 inhibitors {Satoh, 2014 #186}{Tuttolomondo, 2024 #242}.

Experiments comparing Gb3 accumulation to GLA deficiency in human macro- and microvascular cardiac ECs (HMiVECs) how that Gb3 loading disrupts several key endothelial pathways, such as eNOS, iNOS, COX-1, and COX-2, while GLA silencing shows no effects. Cardiac microvascular ECs are more susceptible to Gb3 loading than macrovascular ECs {Namdar, 2012 #241}These data indicate that endothelial dysfunction in Fabry disease is primarily due to Gb3 accumulation rather than GLA deficiency. Gb3 accumulation affects microvascular endothelial cells more severely, altering their phenotype to a vasoconstrictive and pro-inflammatory state, consistent with previous observations that atherosclerosis is not a common finding in Fabry disease {Rombach, 2010 #79}

Gb3 enhances COX-2 expression in HMiVECs, while COX-1 expression remains unaffected, suggesting that vasoconstrictive COX-2 derivatives contribute to microvascular dysfunction in Fabry disease {Tang, 2007 #253}{Park, 2008 #246}. Moreover, Gb3 upregulates VCAM-1 but not ICAM-1 expression in HMiVECs, indicating a pro-inflammatory phenotype in Fabry disease. Increased plasma levels of other endothelial atherosclerotic factors and leukocyte adhesion molecules further support these findings {Shen, 2008 #121}{Demuth, 2002 #254}.

The inflammatory environment induced by Gb3 promotes adhesion molecule expression, such as ICAM-1 and VCAM-1, facilitating leukocyte adhesion and migration to the endothelium. This enhances vascular inflammation and oxidative stress, Studies have shown that Gb3-treated endothelial cells exhibit significantly higher levels of these adhesion molecules, which are critical in the pathogenesis of vascular complications in Fabry disease{Shen, 2008 #121}

Additionally, Gb3 stimulates vascular smooth muscle cell (SMC) proliferation via signaling pathways, including TGF-β and NF-κB, leading to vascular remodeling and stiffness {Barbey, 2006 #171}.The activation of these pathways leads to increased SMC proliferation, contributing to vascular remodeling and stiffness. Hwang et al. (2023) demonstrated that Gb3 also induces autophagy-dependent regulation of necroptosis in endothelial cells, indirectly supporting SMC growth and vascular remodeling{Hwang, 2023 #243}

A novel mechanism by which Gb3 impairs endothelial function involves the degradation of KCa3.1 channels through clathrin-mediated endocytosis and lysosomal degradation, disrupting calcium signaling and NO synthesis {Choi, 2015 #157}.

Gb3 also induces various angiogenesis factors, such as VEGF, VEGFR2, TGF-β, and FGF-2, in endothelial and smooth muscle cells, contributing to vascular dysfunction and angiogenesis in Fabry disease, observed in both in vitro and in vivo models, indicating its role in vascular dysfunction and angiogenesis in Fabry disease {Lund, 2024 #244} {Gambardella, 2023 #240} {Lee, 2012 #245}{Namdar, 2012 #241}{Shen, 2008 #121}{Park, 2008 #246}{Bergmann, 2011 #247}{Loso, 2018 #77}

Research by De Francesco et al. shows that PBMCs from Fabry patients exhibit higher proinflammatory cytokine expression and production when cultured, mediated by TLR4, indicating a proinflammatory response through TLR4 activation {De Francesco, 2013 #55}.

Pollmann et al. highlight that endothelial dysfunction in Fabry disease is linked to glycocalyx degradation, with AGAL-deficient endothelial cells showing reduced glycocalyx and increased monocyte adhesion. Increased angiopoietin-2, heparanase, and NF-κB expression improved with enzyme replacement therapy and anti-inflammatory drugs.{Pollmann, 2021 #17}

  1. Although the authors discuss the severity of the disease in women a small statement on Gender based differences and role of sex hormones in influencing AAFD severity would add to the review.

Thank you for the comment. We added a dedicated paragraph as suggested.

“Gender Differences and the Role of Sex Hormones in the Severity of Fabry Disease

Phenotypic diversity in female AFD patients is primarily due to genetic factors, including XCI and epigenetic modifications. {Sciarra, 2023 #237}{Migliore, 2021 #238}. The variability in female AFD disease severity is linked to X-chromosome inactivation (XCI), where one of the two X chromosomes in females is randomly inactivated, creating a mosaic of affected cells: emales typically carry heterozygous GLA gene mutations, with a 50% chance of passing the defective gene to offspring {Biagini, 2012 #218}{Migliore, 2021 #238}.

XCI can cause varied α-galactosidase A activity, leading to diverse symptoms that appear later and progress more slowly in females compared to males {Migeon, 2020 #214}{Faro, 2023 #38}{Izhar, 2024 #249}. Female AFD patients show clinical pictures, influenced by XCI skewness favoring the mutant allele {Beck, 2019 #215}{Elstein, 2012 #250}. Some studies correlate AFD severity with XCI patterns, while others do not {Beck, 2019 #215}{Elstein, 2012 #250}. Additional factors, such as allele-specific DNA methylation at the GLA promoter, regulate gene expression and contributes to phenotype variability and may influence disease expression and severity and lysoGb3 accumulation (Hossain et al., 2020).

While sex hormones' impact on AFD severity is hypothesized, it remains unproven. Estrogens might offer vascular protection in premenopausal women, but this effect is not definitively established.

Elevated growth factors like VEGF-A and FGF2 in female patients suggest gender-specific mechanisms {Ivanova, 2023 #4}

Recent findings highlight proteins involved in inflammation and coagulation/fibrinolysis, such as ANT3, HRG, FINC, and PLMN, in women {López-Valverde, 2024 #230}. In men, the protein 14-3-3 zeta is significant. Both sexes show complement system activation, indicated by downregulation of C1QB and CO5. Vitronectin expression is downregulated in male AFD patients and in female patients with complications, linking it to atherosclerotic cardiovascular disease pathogenesis, serving as a biomarker for AFD progression.

Understanding these mechanisms can aid in developing targeted therapies and improving patient outcomes.

Reviewer 2 Report

Comments and Suggestions for Authors

Thank you for the opportunity to review this manuscript.

I think that is an important review about the different aspects of FD physiopathology by an expert group in FD. Congratulate for image composition using artificial intelligence, which facilitates comprehension.

Comments:

On line 328 which means "Caceres".

It would be appreciated if each section were numbered.

It would be appreciated if a table could be provided to include in the different sections established in the description the factors included in each section, it would help to understand the described molecules involved in the pathophysiological mechanisms.

On endothelial damage and the risk of thromboembolic problems, we could also add and comment on the study conducted by Kaneski CR, Moore DF, Ries M, Zirzow GC, Schiffmann R. Myeloperoxidase predicts risk of vasculopathic events in hemizgygous males with Fabry disease. Neurology. 2006 Dec 12;67(11):2045-7. doi: 10.1212/01.wnl.0000247278.88077.09

In the section "A Look at Future Perspectives in Therapy, Biomarkers, Molecular Targets, and Mechanisms"  more could be done on the unresolved problems with the available therapies and clinical studies in progress to improve therapeutic results, this section is a bit poor in comparison with the previous ones.

Author Response

Dear Reviewer 2,

We thank you for your thorough review of our manuscript. We have revised our manuscript

according to your helpful suggestions. Each comment is reproduced below in bold followed by

our response. A tracked version of the revision with the changes highlighted has been uploaded.

Thank you for the opportunity to review this manuscript. I think that is an important review about the different aspects of FD physiopathology by an expert group in FD. Congratulate for image composition using artificial intelligence, which facilitates comprehension.

Thank you, we are really glad that our work has been appreciated.

Comments:

  1. On line 328 which means "Caceres".

Sorry, it was a note for bibliography, we removed it and put the reference istead.

  1. It would be appreciated if each section were numbered.

Thank you for the suggestion. We numbered the sections as requested.

  1. It would be appreciated if a table could be provided to include in the different sections established in the description the factors included in each section, it would help to understand the described molecules involved in the pathophysiological mechanisms

Mediators

Role in Vascular Damage/Fibrosis

Clinical Implications

Mechanism of Action

Inflammatory Cells Involved

Gb3/LysoGB3

Accumulation in endothelial cells, disrupts eNOS, reduces NO production, increases ROS production

Endothelial dysfunction, oxidative stress, chronic inflammation, multi-organ involvement

Direct accumulation in cells; potent inflammatory mediator

ECs, SMCs

eNOS

Uncoupling due to Gb3 accumulation, produces superoxide instead of NO; reduced NO bioavailability due to Gb3 accumulation and eNOS uncoupling

Oxidative stress, endothelial damage, increased ROS production  impaired vasodilation, increased risk of thrombosis

Enzyme dysfunction

ECs

ROS

Increased production due to eNOS uncoupling and RAS activation, causes oxidative stress

Oxidative stress, chronic inflammation, tissue damage

Oxidative damage

ECs, SMCs, neutrophils

ICAM-1, VCAM-1

Promote leukocyte adhesion and infiltration, driving chronic inflammation

Vascular inflammation, progression of endothelial damage

Increased expression

ECs, leukocytes

TNF-α, IL6

chronic inflammation, endothelial dysfunction

Increased cardiovascular risk, disease progression

Cytokine signaling

Macrophages, T cells, ECs, SMCs

Mitochondrial Dysfunction

Secondary to lysosomal dysfunction, affects metabolic homeostasis

Altered metabolism and energy, cellular dysfunction, cell death

Disrupted metabolic pathways

ECs, SMCs

CRP

Nonspecific marker of chronic low-grade systemic inflammation, induces cytokines production (IL6, TNF-alfa, IL1), reduces eNOS activity and NO-mediated vasodilation

Indicates ongoing vascular injury and inflammation, prognostic value for CV events

Inflammatory biomarker

Macrophages, hepatocytes, ECs, SMCs, lymphocytes

VEGF

Promotes angiogenesis and endothelial dysfunction

Associated with disease severity and endothelial dysfunction

specific endothelial cell mitogen

ECs

MPO

Elevated in response to oxidative stress, contributes to vascular inflammation

Associated with oxidative stress, vascular inflammation, and endothelial dysfunction, predictor of acute CV events, accelerated atherosclerosis and coronary stenosis,

oxidation of LDL, reduction in NO bioavailability leading to endothelial dysfunction, activation of MMPs

Neutrophils

C3a and C5a

Increased in response to inflammation, contribute to endothelial damage and chronic inflammation

Associated with renal damage and inflammation

influence lymphocyte activity, promoting proliferation and differentiation, recruitment and activation of dendritic cells, cross-talk with TLR, proinflammatory cytokine induction stimulating profibrotic pathways

Macrophages, neutrophils

Syndecan-1

Elevated in endothelial damage, reflects glycocalyx degradation

Indicator of vascular damage, correlates with disease severity and cardiac and renal involvement (heart failure and fibrosis)

Glycocalyx component

ECs

TGF-β1

Promotes fibrosis

associated with renal and cardiac damage, LVH, fibrosis and disease progression

profibrotic cytokine/growth factor: promotes fibrosis in response to chronic inflammation by enhancing the synthesis of ECM

Macrophages, FBs, ECs

FGF2

Elevated levels promote fibrosis and chronic inflammation, particularly in cardiac tissue

Associated with myocardial fibrosis and adverse CV events

Cytokine signaling (regulates angiogenesis, cell growth, and tissue repair)

FBs, ECs

miRNAs

Elevated levels associated with endothelial dysfunction and inflammation

Potential biomarkers for disease monitoring

Post-transcriptional regulation

ECs

IL-10

Anti-inflammatory cytokine,

Modulates inflammatory response, potential marker of disease activity

Cytokine signaling; elevated levels suggest an attempt to counteract inflammation

Macrophages, T cells, ECs

GDF-15

Elevated in response to inflammation and oxidative stress

Indicator of disease severity and progression (kidney injury and cardiovascular involvement and outcomes)

Stress response signaling; modulates renal and cardiac injury, possibly providing protection from tissue injury and fibrosis

Macrophages, ECs

MMP-2 and MMP-9

Indicate extracellular matrix remodeling

Associated with fibrosis and vascular damage (also cardiac and renal)

Extracellular matrix degradation

Macrophages, FBs

Table 2: Inflammatory biomarkers/mediators involved in the pathophysiological mechanisms of vascular damage in Fabry disease.

Abbreviations: ECs, Endothelial cells; SMCs,  smooth muscle cells Gb3, Globotriaosylceramide; NO, Nitric Oxide; eNOS, Endothelial NOS; ROS , Reactive Oxygen Species; ICAM-1, Intercellular Adhesion Molecule 1, VCAM-1 (Vascular Cell Adhesion Molecule 1; TNF-α, Tumor Necrosis Factor Alpha; IL-6, Interleukin 6;  CRP, C-Reactive Protein; VEGF, Vascular Endothelial Growth Factor; MPO, Myeloperoxidase; TLR, toll like receptors; FBs, fibroblasts; TGF-β1, Transforming Growth Factor Beta 1; LVH, left ventricular hypertrophy; ECM, extracellular matrix;  miRNAs, MicroRNAs; FGF2, Fibroblast Growth Factor 2; IL-10, Interleukin 10; GDF-15, Growth Differentiation Factor 15; MMP-2 and MMP-9, Matrix Metalloproteinases.

  1. On endothelial damage and the risk of thromboembolic problems, we could also add and comment on the study conducted byKaneski CR, Moore DF, Ries M, Zirzow GC, Schiffmann R. Myeloperoxidase predicts risk of vasculopathic events in hemizgygous males with Fabry disease. Neurology. 2006 Dec 12;67(11):2045-7. doi: 10.1212/01.wnl.0000247278.88077.09

Elevated levels of myeloperoxidase (MPO) in the blood are significant indicators of acute cardiovascular events, atherosclerosis, coronary stenosis, and endothelial dysfunction, furthering the formation of atherosclerotic plaques by promoting lipid peroxidation and negatively impacting left ventricular function. In their study, Kaneski et al. found that male patients with AFD exhibit significantly higher MPO levels compared to controls, while female heterozygotes also show elevated levels, though not statistically significant{Kaneski, 2006 #212}. Elevated MPO levels correlate with a higher risk of vascular events over the years, and ERT failed to reduce MPO levels even after 55 months of treatment, highlighting a persistent issue. The exact mechanism behind elevated MPO in Fabry disease is not entirely understood. MPO is produced by neutrophils, monocytes, and macrophages in response to reactive oxygen species (ROS), suggesting that neutrophils are particularly activated in Fabry disease, enhancing inflammatory interactions and leukocyte adhesion to endothelial cells. Gb3 also contributes to cell adhesion and B cell apoptosis. The persistent elevation of MPO despite ERT suggests the need for new therapies to reduce vascular events in Fabry disease.

  1. In the section "A Look at Future Perspectives in Therapy, Biomarkers, Molecular Targets, and Mechanisms"  more could be done on the unresolved problems with the available therapies and clinical studies in progress to improve therapeutic results, this section is a bit poor in comparison with the previous ones

Thanks for the suggestion. We have improved and expanded the section by going deeper into unresolved problems with the available therapies and the new therapeutic approaches currently being tested. Specifically, we have included a detailed discussion on the limitations of current treatments and the ongoing clinical trials aimed at enhancing therapeutic outcomes. This enhancement has enriched the section, making it more comprehensive and aligned with the depth of the previous sections. We added these paragraphs: 

“AFD management remains challenging despite advancements in ERT. While ERT has demonstrated efficacy in improving organ function, it requires biweekly intravenous administration, which poses risks such as immunogenicity that can ultimately reduce its effectiveness {Lerario, 2024 #219}. Anti-drug antibodies (ADAs) often develop within three to six months after starting ERT, predominantly affecting males with classic AFD. These antibodies can bind to the therapeutic enzyme, leading to its inactivation and reduced cellular uptake, thus diminishing ERT efficacy and worsening clinical outcomes {Lenders, 2018 #251}{Lerario, 2024 #219}. Strategies to mitigate these effects, such as increasing ERT dosage and using immunosuppressive therapies, have shown varied long-term effects and potential side effects {Lenders, 2018 #251};NCT03614234).

Despite these challenges, significant progress is being made in developing new therapeutic approaches for Fabry disease. Ongoing research and clinical trials are focused on more effective and less burdensome treatments, including oral therapies, novel ERT formulations, gene therapy, and substrate reduction therapy (SRT).

One promising oral therapy is migalastat, a pharmacological chaperone that stabilizes the α-GalA enzyme, facilitating its proper trafficking and function. Clinical trials, such as FACETS and ATTRACT {Germain, 2016 #259}{Hughes, 2017 #48} have demonstrated its efficacy in reducing left ventricular mass index and stabilizing eGFR. Substrate reduction therapies are another exciting area of research. Drugs like lucerastat and venglustat inhibit glycosphingolipid synthesis, aiming to reduce Gb3 accumulation in tissues. These therapies are being tested for their potential to modify the disease course and improve symptoms, offering a new avenue for patients with non-classical mutations or those who could benefit from a combined approach with ERT (MODIFY trial, NCT03425539; NCT0528054; NCT05206773).

Lucerastat, an orally active glucosylceramide synthase (GCS) inhibitor, aims to restore enzyme activity without intravenous administration by blocking the synthesis of a precursor molecule of Gb3. Phase 1 trial results (NCT02930655) showed lucerastat was well tolerated and significantly reduced blood levels of Gb3 and related fatty molecules compared to ERT alone, though no effects were observed on kidney or heart function. Most participants opted to continue in an open-label extension study (NCT03737214), where interim results for up to two years showed maintained reductions in Gb3. While kidney function had been declining in participants before entering MODIFY, data from the Phase 3 trial and its extension study showed lucerastat was associated with slower declines compared with a placebo and historical data. Improvements in cardiac health measures were also observed, including a decrease in the left ventricular mass index, a predictor of cardiovascular disease.

Venglustat, another SRT, is being evaluated in the CARAT trial (NCT05280548) for its effect versus standard therapy on left ventricular mass at CMR and organ damage in Fabry disease patients with LVH.

AL01211, an investigational therapy, has shown promise in Phase 1 trials by safely lowering levels of Gb3 in healthy adults. This oral treatment is currently in a Phase 2 trial (NCT06114329) involving men with classic Fabry disease who have never used a disease-related treatment. AL01211 aims to maximize its effects on targeted organs and minimize brain side effects by avoiding the central nervous system.

In addition to oral therapies, new formulations of ERT are being explored. Pegunigalsidase alfa, a PEGylated ERT, has shown sustained plasma concentrations and efficacy  in reducing plasma lyso-Gb3 and its deposition in the kidney (slowing eGFR slope and proteinuria) and in the left ventricle, potentially offering a less immunogenic alternative to current ERT options., as PEGylation could mask some epitopes from the immune system, reverse anti-drug antibodies and induce immune tolerance {Schiffmann, 2019 #260}{Holida, 2019 #261} {Wallace, 2024 #262}{Linhart, 2023 #263}. Ongoing clinical trials (NCT03614234, NCT06095713, NCT05710692, NCT03566017) will evaluate its efficacy and safety also in real world compared to previously approved ERTs. Another innovative approach involves a plant-based ERT derived from Phycomitrella patens, designed to elude immunogenic responses, that has shown to reduce urinary Gb3 levels{Shen, 2016 #264}{Hennermann, 2019 #265}.

Gene therapy holds significant promise for Fabry disease, with several approaches under investigation. One method uses adeno-associated virus (AAV) vectors for one-time intravenous administration to deliver the GLA gene to hepatocytes, enabling endogenous enzyme production and secretion. Clinical trials (NCT06270316, NCT04519749, STAAR - NCT04046224) are exploring the safety and efficacy of these therapies. Another approach involves lentivirus-mediated gene therapy, using lentivirus-transduced hematopoietic stem/progenitor cells to express α-galactosidase A. Early results are promising, showing increased enzyme activity and reduced Gb3 and lyso-Gb3 levels (NCT02800070). Additionally, mRNA therapy, encapsulating GLA mRNA in lipid nanoparticles, has shown potential in preclinical studies to increase enzyme levels in the liver, heart, and kidneys.

Moreover, combination therapies and novel administration protocols are being investigated to enhance the effectiveness of current treatments. The SHORTEN study (NCT06019728) aims to optimize ERT infusion protocols to reduce treatment burden by increasing the infusion rate and reducing infusion volume. The RECAFTURE trial (NCT02469181) evaluates the impact of ERT on cardiac function using advanced imaging techniques to understand its effects on left ventricular diastolic function and flow.

Other ongoing trials in Fabry Disease focus on the role of inflammation in diagnosis and its correlation with imaging. The study NCT06226987 combines PET and MRI to identify myocardial inflammation, distinguishing between edema and fibrosis. The Bio-FAIR trial (NCT06007768) examines immune response biomarkers and their relationship with disease severity, quality of life, and pain in patients. The study NCT05698901 aims to develop an algorithm using biomarkers to evaluate disease progression and treatment response. Lastly, the FASHION study (NCT05761834) explores differences in inflammatory profiles between Fabry Disease and hypertrophic cardiomyopathy, analyzing correlations between inflammatory phenotypes and cardiac severity through proteomic and transcriptomic analyses, echocardiography, and electrocardiography.

The future of Fabry disease treatment looks promising, with these innovative approaches potentially improving patient outcomes and quality of life”.

Reviewer 3 Report

Comments and Suggestions for Authors

Authors provided an interesting review about dedicated mechanisms supporting pathological consequences in AFD.

The review is pretty easy to read, with several crucial issues adequately engaged.

Because of the several cited immunity and autoimmunity, I would suggest to include dedicated paragraphs or full chapter describing, comprehensively, which immune cells are involved and their place and interplay with damaged endothelium and inflammation.

Moreover, several mechanisms are shared with atherosclerosis, but specimen of pathological findings in AFD are different: these differences should be underscore in order to enlighten the difference between the diseases (especially for chapter on nitric oxide and inflammation).

I would suggest authors to consider moving chapter on nitric oxide between others involving pathophysiological mechanisms: endothelial dysfunction and inflammation, instead between vascular and myocardial alterations.

Figures:

- Figure 2 is a bit confused, merging molecular, radiological, ultrasound and clinical findings together with detrimental consequences without a clear rationale and logical schema. It should be improved.

- In the Figure 1, the box placed in the panel A partially refers to panel B. Moreover, the statement that biomarkers correlate with MSSI score is misleading: in the text, authors only report IL-6 levels, and TNF-alfa only in treated patients.

Minor:

- lines 23: "and" instead "cand".

- please be consistent in the use of acronyms (i.e.: GL-3 and Gb3).

- please fix bibliography (i.e. lines 176, 181, etc.).

- check capital letters (i.e. lined 475).

Author Response

Dear Reviewer 3,

We thank you for your thorough review of our manuscript. We have revised our manuscript

according to your helpful suggestions. Each comment is reproduced below in bold followed by

our response. A tracked version of the revision with the changes highlighted has been uploaded.

Authors provided an interesting review about dedicated mechanisms supporting pathological consequences in AFD. The review is pretty easy to read, with several crucial issues adequately engaged.

Thanks for the positive and appreciative comments.

Because of the several cited immunity and autoimmunity, I would suggest to include dedicated paragraphs or full chapter describing, comprehensively, which immune cells are involved and their place and interplay with damaged endothelium and inflammation.

Thanks for the suggestion, we added a dedicated paragraph to the manuscript.

“Immune Cells Involved in Fabry Disease and Their Interplay with Endothelium and Inflammation”

The accumulation of Gb3 in various tissues results in significant endothelial dysfunction and inflammation, where the immune system plays a critical role.

Neutrophils have a leading role in AFD-related inflammation. These cells produce myeloperoxidase (MPO), an enzyme that is significantly elevated in AFD patients and is closely associated with endothelial dysfunction and increased cardiovascular risk. Elevated MPO levels indicate a state of neutrophil activation, which generates reactive oxygen species (ROS), exacerbating oxidative stress and endothelial damage. This oxidative stress contributes to a vicious cycle of inflammation and vascular injury {Kaneski, 2006 #212}. Monocytes and macrophages are essential in the inflammatory response in AFD. Monocytes differentiate into macrophages, which produce various pro-inflammatory cytokines and enzymes, including MPO. These macrophages accumulate in tissues where Gb3 is deposited, perpetuating chronic inflammation and tissue remodeling. The persistent activation of these cells leads to enhanced inflammatory responses and further endothelial damage {Kurdi, 2024 #248}{Rozenfeld, 2017 #11}. T cells, particularly the Th1 and Th17 subsets, contribute to the immune response in AFD. Th1 cells produce interferon-gamma, while Th17 cells secrete IL-17, both of which promote inflammation. These cytokines are implicated in the vascular dysfunction observed in AFD, suggesting that T cells play a significant role in sustaining chronic inflammation and endothelial injury {Rozenfeld, 2017 #11). B cells also have a critical role in AFD, contributing to its autoimmune component. These cells produce antibodies against Gb3 and other autoantigens, leading to immune complex formation, which can deposit in blood vessels, exacerbating endothelial damage and inflammation.

The accumulation of Gb3 in endothelial cells triggers a cascade of inflammatory responses: it activates Gb3 and lyso-Gb3 accumulation activates TLRs and inflammasomes, primarily stimulating the innate immune response. The adaptive immune system also plays a role, with Natural killer (NK) T cells recognizing self-neutral glycosphingolipids as antigens. This inflammatory cascade involves various cytokines and chemokines, contributing to chronic inflammation and fibrosis in affected organs, underscoring the need for improved therapeutic interventions. Invariant NK cells, which have characteristics of both NK cells of the innate immune system and T cells of the adaptive immune system, recognize glycolipids presented by CD1d molecules and contribute to the immune response in AFD. Dendritic cells also play a dual role in both innate and adaptive immune responses, further highlighting the interconnected nature of these immune pathways in AFD {Kurdi, 2024 #248}.

One of the first responses is the upregulation of adhesion molecules such as VCAM-1 and ICAM-1 on the surface of endothelial cells. These molecules facilitate the adhesion and transmigration of immune cells into the vascular endothelium, promoting further inflammation and endothelial damage{Kurdi, 2024 #248}{Shen, 2008 #121}. Endothelial activation in AFD is characterized by a shift towards a pro-inflammatory and pro-thrombotic state. Activated endothelial cells secrete cytokines and chemokines, attracting more immune cells and perpetuating the cycle of inflammation. This chronic inflammatory state leads to vascular remodeling, including intimal hyperplasia and fibrosis, contributing to the narrowing of blood vessels and impaired blood flow{Rozenfeld, 2017 #11}. Oxidative stress plays a crucial role in AFD-related endothelial dysfunction. MPO released by neutrophils generates ROS, which cause direct oxidative damage to endothelial cells. This oxidative stress not only damages the endothelium but also amplifies the inflammatory response, creating a self-perpetuating cycle of vascular pathology.

Moreover, several mechanisms are shared with atherosclerosis, but specimen of pathological findings in AFD are different: these differences should be underscore in order to enlighten the difference between the diseases (especially for chapter on nitric oxide and inflammation).

Thanks for the suggestion, we added a dedicated paragraph to the manuscript.

“Pathophysiological Differences between Fabry Disease and Atherosclerosis”

AFD and atherosclerosis both involve endothelial dysfunction and inflammation, but their pathophysiological mechanisms differ significantly. AFD's vascular damage is driven by the accumulation of Gb3, leading to oxidative stress, impaired nitric oxide (NO) synthesis, and a specific inflammatory response triggered by Gb3 accumulation. This accumulation disrupts endothelial nitric oxide synthase (eNOS) function, causing eNOS uncoupling and the formation of superoxide instead of NO, leading to oxidative stress and vascular inflammation {Rozenfeld, 2017 #11}{Stamerra, 2021 #1}. Additionally, the upregulation of adhesion molecules and the activation of the local renin-angiotensin system further exacerbate vascular damage and inflammation {Rozenfeld, 2017 #11}{Stamerra, 2021 #1}. Unlike atherosclerosis, AFD rarely leads to obstructive coronary artery disease (CAD). Instead, patients exhibit increased intima-media thickness (IMT) in carotid arteries and coronary microvascular dysfunction without significant plaque formation, leading to non-obstructive myocardial infarctions {Ivanova, 2023 #4} {Graziani, 2022 #266}

In contrast, atherosclerosis is driven by lipid accumulation and immune responses within the arterial walls. The process begins with the deposition of oxidized low-density lipoproteins (oxLDL) in the arterial intima, leading to endothelial activation and the recruitment of monocytes. These monocytes differentiate into macrophages, engulf lipids, and form foam cells, initiating plaque formation {Bäck, 2019 #268}{Wolf, 2019 #269}. Persistent inflammation involves various immune cells, including T cells, dendritic cells, and macrophages, which secrete pro-inflammatory cytokines (e.g., TNF-α, IL-6) and contribute to plaque growth and instability{Paulson, 2010 #270}. Atherosclerotic plaques can rupture, leading to thrombus formation and acute cardiovascular events such as myocardial infarction and stroke{Wu, 2017 #272}. Atherosclerosis is strongly associated with traditional cardiovascular risk factors, including hypercholesterolemia, hypertension, diabetes, smoking, and elevated levels of lipoprotein(a) and uric acid, which exacerbate endothelial dysfunction and promote plaque development {Saigusa, 2020 #271}{Bäck, 2019 #268}

In terms of clinical presentation, AFD typically does not feature flow-limiting plaques but rather diffuse intima-media thickening (IMT) and microvascular dysfunction. AFD patients exhibit increased IMT in carotid and coronary arteries without significant atheromatous plaques, leading to myocardial infarctions not caused by occlusive coronary artery disease but rather by microvascular dysfunction and endothelial impairment. This is evident in studies demonstrating that Fabry patients have higher IMT but no significant coronary artery stenosis, with myocardial infarctions being type 2 rather than type 1 {Stamerra, 2021 #1}. To synthesize the main differences, we can focus on the following points:

- Molecular Pathways: In AFD, the primary driver is the accumulation of Gb3, leading to oxidative stress and inflammation, whereas in atherosclerosis, lipid accumulation and subsequent chronic inflammation are central.

- Immune Cells and Cytokines: AFD involves a specific inflammatory response triggered by Gb3 accumulation, whereas atherosclerosis involves a broader range of immune cells and cytokines associated with lipid-induced inflammation.

- Clinical Manifestations: AFD typically does not result in obstructive CAD but is associated with microvascular dysfunction and non-obstructive myocardial infarctions. Atherosclerosis, in contrast, leads to plaque formation and obstructive cardiovascular events.

Understanding these differences is important for developing targeted therapeutic strategies. In AFD, treatments focus on reducing Gb3 accumulation and addressing its metabolic consequences, whereas in atherosclerosis, therapies aim to manage lipid levels, control inflammation, and stabilize plaques.

I would suggest authors to consider moving chapter on nitric oxide between others involving pathophysiological mechanisms: endothelial dysfunction and inflammation, instead between vascular and myocardial alterations.

Thank you for your valuable suggestion. We have revised the manuscript and moved the chapter on nitric oxide to the section discussing pathophysiological mechanisms, specifically before  endothelial dysfunction.

Figures:

- Figure 2 is a bit confused, merging molecular, radiological, ultrasound and clinical findings together with detrimental consequences without a clear rationale and logical schema. It should be improved.

Thank you for your valuable feedback. We have addressed your concerns regarding Figure 2 with the following changes:

  1. We have maintained the pathogenetic bases at the top of the figure.
  2. We have eliminated the radiological and imaging components.
  3. We have preserved the pathophysiological aspects along with the related clinical manifestations.

These modifications aim to provide a clearer rationale and logical schema to improve the overall clarity of Figure 2.

- In the Figure 1, the box placed in the panel A partially refers to panel B. Moreover, the statement that biomarkers correlate with MSSI score is misleading: in the text, authors only report IL-6 levels, and TNF-alfa only in treated patients.

Thank you for your comments and suggestions. We have made the following changes to improve the clarity and accuracy of our work:

We have removed the misleading statement and modified the declaration to be less assertive, as follows: "IL-6 and TNF-alpha correlate with disease severity and progression in ERT-treated patients.

It was our intention to create continuity between panel A and panel B. Therefore, we removed the division between panel A and panel B and connected the two image panels with an arrow to create the link.

Minor:

  1. lines 23: "and" instead "cand"

Thanks for the suggestion, we fixed this issue

  1. please be consistent in the use of acronyms (i.e.: GL-3 and Gb3).

Thanks for the suggestion, we fixed this issue, keeping the acronym Gb3 in all cases.

  1. please fix bibliography (i.e. lines 176, 181, etc.).

Thanks for the suggestion, we fixed this issue

  1. check capital letters (i.e. lined 475).

Thanks for the suggestion, we fixed this issue